# Comparison of freshly cultured versus cryopreserved mesenchymal stem cells in animal models of inflammation: A pre-clinical systematic review

Chintan Dave[1], Shirley HJ Mei[2], Andrea McRae[2], Christine Hum[3,4], Katrina J Sullivan[3], Josee Champagne[3,5], Tim Ramsay[5], Lauralyn McIntyre[3,6]*

[1]Division of Critical Care Medicine, Department of Medicine, Western University, London, Canada; [2]Regenerative Medicine Program, Ottawa Hospital Research Institute, Ottawa, Canada; [3]Knowledge Synthesis Group, Ottawa Hospital Research Institute, Ottawa, Canada; [4]University of Ottawa, Ottawa, Canada; [5]Clinical Epidemiology, Ottawa Hospital Research Institute, Ottawa, Canada; [6]Division of Critical Care, Department of Medicine, University of Ottawa, Ottawa, Canada

*For correspondence: lmcintyre@ohri.ca

Competing interest: The authors declare that no competing interests exist.

## Abstract

**Background:** Mesenchymal stem cells (MSCs) are multipotent cells that demonstrate therapeutic potential for the treatment of acute and chronic inflammatory-mediated conditions. Although controversial, some studies suggest that MSCs may lose their functionality with cryopreservation which could render them non-efficacious. Hence, we conducted a systematic review of comparative pre-clinical models of inflammation to determine if there are differences in in vivo measures of pre-clinical efficacy (primary outcomes) and in vitro potency (secondary outcomes) between freshly cultured and cryopreserved MSCs.

**Methods:** A systematic search on OvidMEDLINE, EMBASE, BIOSIS, and Web of Science (until January 13, 2022) was conducted. The primary outcome included measures of in vivo pre-clinical efficacy; secondary outcomes included measures of in vitro MSC potency. Risk of bias was assessed by the SYRCLE 'Risk of Bias' assessment tool for pre-clinical studies.

**Results:** Eighteen studies were included. A total of 257 in vivo pre-clinical efficacy experiments represented 101 distinct outcome measures. Of these outcomes, 2.3% (6/257) were significantly different at the 0.05 level or less; 2 favoured freshly cultured and 4 favoured cryopreserved MSCs. A total of 68 in vitro experiments represented 32 different potency measures; 13% (9/68) of the experiments were significantly different at the 0.05 level or less, with seven experiments favouring freshly cultured MSC and two favouring cryopreserved MSCs.

**Conclusions:** The majority of preclinical primary in vivo efficacy and secondary in vitro potency outcomes were not significantly different (p<0.05) between freshly cultured and cryopreserved MSCs. Our systematic summary of the current evidence base may provide MSC basic and clinical research scientists additional rationale for considering a cryopreserved MSC product in their pre-clinical studies and clinical trials as well as help identify research gaps and guide future related research.

**Funding:** Ontario Institute for Regenerative Medicine

## Editor's evaluation

The pre-clinical systematic review by Dave C et al. covers an important and highly debated topic, which is the advantages and disadvantages of the use of freshly cultured vs cryopreserved mesenchymal stromal cells (MSCs). The authors conduct an appropriate survey and bias analysis and focus their review on reported studies on animal models of inflammation. They conclude that there are no significant differences between freshly-isolated or cryopreserved MSCs in terms of their pre-clinical efficacy.

## Introduction

Mesenchymal stromal cells (mesenchymal stem cells; MSCs) are multipotent stem cells that can be isolated from many adult tissues (e.g. bone marrow, adipose tissue) (*Pittenger et al., 2019*). MSCs have been studied in clinical trials for almost two decades (*Koç et al., 2000*), and have since been implicated in use for diverse conditions (*Gomez-Salazar et al., 2020*). MSCs release growth factors and cytokines along with extracellular vesicles to activate cell proliferation, prevent apoptosis, and ultimately improve regenerative response (*Pittenger et al., 2019*). MSCs may also modulate the immune response by decreasing inflammation, reducing scar formation, increasing pathogen clearance, altering endothelial permeability, and improving mitochondrial dysfunction as demonstrated in different pre-clinical models of inflammation (*Fish and Hajjar, 2015*; *Hoogduijn et al., 2010*; *Gupta et al., 2012*; *Islam et al., 2012*; *Li et al., 2018*; *Tsubokawa et al., 2010*). The mechanism for how MSCs modulate inflammation and promote healing is not yet completely understood; however, the observed effect may be mediated by both the direct contact with immune cells and release of soluble factors (*Caplan, 2009*; *Shi et al., 2012*; *Souza-Moreira et al., 2022*). Given their potent immuno-modulatory effects, MSCs are particularly attractive for use in infectious as well as acute and chronic inflammatory conditions. There are a growing number of studies that demonstrate the efficacy of MSC therapy in a variety of pre-clinical models, such as acute lung injury (*Chang et al., 2014*; *Mei et al., 2007*; *Matthay et al., 2010*; *Weiss et al., 2013*; *Wilson et al., 2015*), sepsis (*McIntyre et al., 2018*; *Mei et al., 2010*), acute myocardial infarction (*Boyle et al., 2010*), multiple sclerosis (*Connick et al.,*

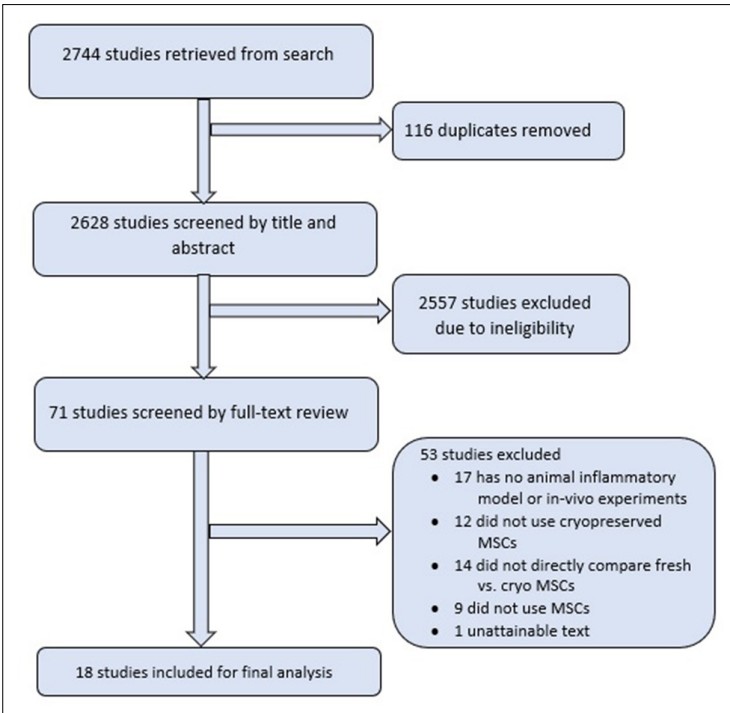

**Figure 1.** Literature search and study inclusion.

*2011*), graft-versus-host disease (*Baron et al., 2010*; *Introna et al., 2014*; *Pérez-Simon et al., 2011*), osteoarthritis (*Emadedin et al., 2015*; *Jo et al., 2014*; *Orozco et al., 2014*; *Vega et al., 2015*; *Vives et al., 2015*), and inflammatory bowel disease (IBD) (*Forbes et al., 2014*; *Molendijk et al., 2015*). Moreover, as of March 10, 2022, 1,097 active trials involving MSCs were registered (https://www.clinicaltrials.gov). Although MSCs have potential to treat many clinical conditions, a major limitation with nearly all studies is the constrained real-world applicability, where it is vital to have an intervention that is readily available and administered in a time-sensitive manner. For this to occur, the MSCs must overcome the logistical challenges of in-vitro isolation and culture, effective cryopreservation methodology, and a route for rapid accessibility to the bedside. Future real-world therapeutic applications of MSCs will need to be ready for immediate use as off-the-shelf products in urgent medical situations (*Mendicino et al., 2014*; *Woods et al., 2016*).

To date, a majority of preclinical MSC research employ freshly cultured MSCs. In a recent systematic review of the safety of MSCs in 55 randomized clinical trials, only 15 (27%) used cryopreserved cells (*Thompson et al., 2020*), potentially due to the concern that cryopreserved MSCs may lose some of their functionality (*Galipeau et al., 2016*). Some in vitro studies demonstrate a negative impact of cryopreservation on MSC function (*François et al., 2012*; *Chinnadurai et al., 2016*); however, others suggest that cryopreservation may not negatively impact their functionality (*Cruz et al., 2015*; *Devaney et al., 2015*; *Gramlich et al., 2016*; *Luetzkendorf et al., 2015*).

To evaluate evidence currently available in the literature, our team conducted a systematic synthesis of all pre-clinical comparative studies that examined freshly cultured versus cryopreserved MSCs on surrogate measures of in vivo efficacy (primary outcomes) and in vitro potency (secondary outcomes) in animal models of inflammation. The protocol for our systematic review is published in *Systematic Reviews* (https://doi.org/10.1186/s13643-020-01437-z) and registered in PROSPERO (CRD42020145833).

## Materials and methods
### Search strategy
We conducted electronic search strategies without language restriction of Ovid platform, Ovid MEDLINE, OvidMEDLINE In-Process & Other Non-Indexed Citations, Embase Classic plus Embase, and BIOSIS and Web of Science using Web of Knowledge until January 13, 2022. Given the non-standard terminology associated with MSCs, several pre-defined terms were used, and the electronic and manual search strategies were developed and tested through an iterative process by an experienced medical information specialist in consultation with the research team (*Supplementary file 1*). Six target articles provided by an expert in the field of preclinical research (SM) that were known prior to the search were included in the search criteria to help capture all potential studies. No additional filters were employed to ensure the largest number of relevant studies are captured. We followed the PRISMA guidelines (*Supplementary file 2*) for reporting our systematic review.

### Assessment of risk of bias
Risk of bias was assessed independently by two reviewers (CD and AM), and disagreements were resolved via consensus, or by a third reviewer when necessary. All studies were assessed as high, low, or unclear for the 10 domains of bias adapted from the SYRCLE 'Risk of Bias' assessment tool for pre-clinical in vivo studies (*Hooijmans et al., 2014*). This tool has been adapted from the Cochrane Collaboration Risk of Bias tool employed in clinical studies, with an aim to incorporate key elements that are relevant for in vivo animal studies. The prompting questions employed to assess risk of bias (AGREE tool) can be found in *Supplementary file 3*. The 10 risk-of-bias domains and signalling questions are provided in Table 7*.*

### Study eligibility
Pre-clinical studies of in vivo models of inflammation that directly compared freshly cultured to cryopreserved MSC products (randomized, quasi-randomized, and non-randomized designs) were included. To be defined as cryopreserved, MSCs could have been cryopreserved for any duration of time and/or be placed in culture for less than 24 hr post-thaw prior to use in the given experiment. MSCs were considered freshly cultured when the cells were either in continuous culture or cryopreserved but then thawed and placed in culture for at least 24 hr prior to use in experiments. We used this 24-hr culture

**Table 1.** Models of inflammation and characteristics of included studies.

| First Author (Year) | Animal Inflammatory Model | Country | Language of Publication | Species | Strain | Gender | Sample size | Age (range) | Weight (grams) |
|---|---|---|---|---|---|---|---|---|---|
| *Bárcia et al., 2017* | 1) Chronic adjuvant-induced arthritis (AIA) model 2) Hindlimb ischemia model | Portugal | English | 1) Rat 2) Mouse | 1) Winstar 2) C57BL/6 | 1) Male 2) Female | 1) 18 2) 36 | 1) NR 2) 12 weeks | 1) 365–480 g 2) NR |
| *Cruz et al., 2015* | Allergic Airways Inflammation induced by Aspergillus hyphal extract (AHE) exposure in immunocompetent mice | USA | English | Mouse | C57BL/6 | Male | 72 | 8–12 weeks | NR |
| *Curley et al., 2017* | Acute respiratory distress syndrome by intratracheal instillation of E. coli | Canada | English | Rat | Sprague-Dawley (specific pathogen-free) | Male | NR | NR | 350–450 g |
| *Devaney et al., 2015* | Acute lung injury induced by E. coli pneumonia | Ireland | English | Rat | Sprague-Dawley (specific pathogen-free) | Male | 40 | NR | 350–450 g |
| *Gramlich et al., 2016* | Retinal Ischemia/Reperfusion Injury Model | USA | English | Mouse | C57BL6/J | Male and Female | 37 | 2 months | NR |
| *Lohan et al., 2018* | Corneal Transplantation | Ireland | English | Rat | Lewis | Male | NR | 8–14 weeks | NR |
| *Salmenkari et al., 2019* | Colitis (3% DSS) | Finland | English | Mouse | Balb/c | Male | NR | 8 weeks | NR |
| *Somal et al., 2017* | Wound healing | India | English | Rat | Wistar | Male | 27 | NR | NR |
| *Bharti et al., 2020* | Wound healing | India | English | Guinea pigs | Dunkin Hartley | Male | 25 | NR | NR |
| *Horie et al., 2020a* | Ventilator-induced Lung Injury | Ireland | English | Rats | NR | NR | NR | NR | NR |
| *Horie et al., 2020a* | E. coli-induced lung injury | Ireland | English | Rats | Pathogen-free sprague Dawley | Male | NR | NR | 300–450 g |
| *Khan et al., 2019* | Spinal Cord Injury induced through a balloon compression method | Korea | English | Dog | Beagle | NR | 12 | 1.2+/-0.2 years | 12+/-3 kg |
| *Rogulska et al., 2019* | Wound healing | Ukraine | English | Mouse | Balb/C | Male | 27 | 5–6 months | 25–30 g |
| *Tan et al., 2019* | Polymicrobial sepsis induced by cecal-ligation-and-puncture (CLP) | Canada | English | Mouse | C57BL6/J | Female | NR | NR | 17–21 g |
| *Perlee et al., 2019* | K.K. pneumoniae induced pneumosepsis | Netherlands | English | Mouse | Pathogen free C57BL/6 | Female | NR | 8–10 weeks | NR |
| *Yea et al., 2020* | Wound healing | Korea | English | Rat | Sprague-Dawley | Male | 120 | 12 weeks | 340–360 g |
| *Horiuchi et al., 2021* | Osteoarthritis | Japan | English | Rat | Wildtype Lewis | Female | 40 | 10 weeks | 180–200 g |
| *Horie et al., 2021* | Ventilator-Induced Lung Injury | Ireland | English | Rat | Sprague-Dawley | Male | 28 | NR | 350–450 g |

NR = Not Reported.

time as a cut-off as previous experiments suggest that cryopreserved MSCs may require 24 hr of culture to recover their functionality (*Galipeau, 2013*). The study must have included an *animal* model of inflammation where the intervention and comparison groups examined the administration of cryopreserved and freshly cultured MSCs, respectively, delivered by any route, and derived from the same MSC origin (ex. bone marrow, adipose tissue, umbilical cord, or other) and source (xenogeneic, syngeneic, autologous, or allogeneic). MSCs that were pre-treated, pre-conditioned, genetically altered, or co-administered with other experimental interventions were included if the same alteration was applied to both the freshly cultured and cryopreserved MSCs.

Studies that administered MSCs before or during the induction of the experimental pre-clinical model (i.e. prevention studies) were excluded. We also excluded studies of immunocompromised animals (SCID) or treatments to immunosuppress the animals were excluded because our primary aim was to examine the efficacy of cryopreserved versus freshly cultured MSCs on measures of inflammation in animal models with an intact immune system. Moreover, an intact immune system may be required for MSC immunomodulation via the host cytotoxic cell activity (*Galleu et al., 2017*). Studies that examined the effects of MSCs on implantation and tissue regeneration (e.g. bone regeneration), or compared differentiated MSCs (e.g. differentiated into a myocyte), Mesenchymal Progenitor Cells (MPCs), Mononuclear Cell (MNC) fraction, or stem cells that were not described as MSCs, and studies that only reported in vitro experiments comparing freshly cultured to cryopreserved MSC products were also excluded.

### Outcomes

The primary outcomes were surrogate measures of in vivo pre-clinical efficacy that were relevant to specific acute and chronic inflammatory animal models and defined by two outcome domains: 1) The Function and Composition of Tissues (e.g. organ dysfunction, histopathological damage); and 2) Protein Expression and Secretion (e.g. cytokine levels, immunohistochemistry analysis).

Secondary outcomes included measures of in vitro MSC potency (that were described as additional experiments in the included in vivo studies). Ideally, potency should represent the MSCs' mechanism of action; however, MSCs have complex and multiple mechanisms of action, all of which are not yet fully characterized or reported (*Galipeau et al., 2016*). In accordance with the International Society for Cellular Therapy perspective paper on this topic (*Galipeau et al., 2016*), MSC potency was based on an assay matrix (collection of assays) that included a combination of in vitro analytical and/or biological assays (e.g. the cellular secretome by ELISA [enzyme-linked immunosorbent assay], or functional cell-based assays [in vitro assay culturing MSCs with responder immune cells] respectively). Hence, the two main secondary in vitro potency outcome domains were: 1) Co-culture assays; and 2) Protein Expression and Secretion (ex: cytokine levels).

### Study selection and data collection

The titles and abstracts were screened independently by two members (CD, ED). The full-text of all potentially eligible studies were retrieved and reviewed for eligibility, independently, by two members of the team using the a priori eligibility criteria described above. Disagreements between reviewers were resolved by consensus or by a third member of the systematic review team (LM, SM). Data were extracted independently by two members of the research team into standardized, pilot-tested excel sheet forms (*Supplementary file 4*). Authors were contacted for data clarification or for additional data when required.

### Data analysis

Meta-analyses were planned as per protocol, if sufficient data were available and if appropriate: two or more studies with similar disease models for an in vivo pre-clinical efficacy outcome, with the same outcome definition. Data reported in non-standard format (e.g. mean ± standard error, median and range) was converted to mean ± standard deviation. Given the complexity and variety of results, the results were summarized in tabular format and presented as number of experiments that reached statistical significance at the 0.05 level.

## Results
### Search results and study characteristics

The search strategy yielded 2744 potential studies; and after applying the eligibility criteria and full text review, 18 studies were deemed eligible for inclusion (*Figure 1*; *Cruz et al., 2015*; *Devaney*

*et al., 2015*; *Gramlich et al., 2016*; *Salmenkari et al., 2019*; *Somal et al., 2017*; *Tan et al., 2019*; *Curley et al., 2017*; *Horiuchi et al., 2021*; *Horie et al., 2021*; *Yea et al., 2020*; *Bárcia et al., 2017*; *Bharti et al., 2020*; *Khan et al., 2019*; *Horie et al., 2020a*; *Lohan et al., 2018*; *Perlee et al., 2019*; *Rogulska et al., 2019*).

Eight studies used mice for their experiments (*Cruz et al., 2015*; *Gramlich et al., 2016*; *Salmenkari et al., 2019*; *Somal et al., 2017*; *Tan et al., 2019*; *Curley et al., 2017*; *Perlee et al., 2019*; *Rogulska et al., 2019*), seven studies used rats (*Devaney et al., 2015*; *Curley et al., 2017*; *Horiuchi et al., 2021*; *Horie et al., 2021*; *Yea et al., 2020*; *Lohan et al., 2018*; *Horie et al., 2020b*), one study used both mice and rats (*Bárcia et al., 2017*), one study used beagle dogs (*Khan et al., 2019*), and one study used guinea pigs (*Bharti et al., 2020*). Twelve studies included a 'vehicle only' as an additional control arm (*Devaney et al., 2015*; *Gramlich et al., 2016*; *Salmenkari et al., 2019*; *Somal et al., 2017*; *Horiuchi et al., 2021*; *Horie et al., 2021*; *Yea et al., 2020*; *Bharti et al., 2020*; *Lohan et al., 2018*; *Perlee et al., 2019*; *Rogulska et al., 2019*; *Horie et al., 2020b*), while four studies employed a sham animal model, where disease negative animals received MSCs or vehicle (*Tan et al., 2019*; *Curley et al., 2017*; *Bárcia et al., 2017*; *Horie et al., 2020a*). One study directly compared cryopreserved and freshly cultured MSCs without an additional control arm (*Khan et al., 2019*) and one study employed a sham model, vehicle, and cryopreserved and freshly cultured fibroblasts as controls (*Cruz et al., 2015*).

Of the 18 included studies, seven studied models of preclinical lung injury and sepsis (*Devaney et al., 2015*; *Tan et al., 2019*; *Curley et al., 2017*; *Horie et al., 2021*; *Horie et al., 2020a*; *Perlee et al., 2019*; *Horie et al., 2020b*), four a wound healing model (*Somal et al., 2017*; *Yea et al., 2020*; *Bharti et al., 2020*; *Rogulska et al., 2019*), three of neurological or ocular disease, specifically one of corneal transplantation (*Lohan et al., 2018*), retinal ischemia/reperfusion (*Gramlich et al., 2016*), and spinal cord injury model (*Khan et al., 2019*), and one each of allergic airway inflammatory disease (*Cruz et al., 2015*), wound healing and chronic inflammatory arthritis (*Bárcia et al., 2017*), acute and chronic inflammatory colitis (*Salmenkari et al., 2019*), and chronic osteoarthritis (*Horiuchi et al., 2021*). Complete reporting of inflammatory models, MSC origins and characteristics can be found in *Tables 1 and 2*.

## Description of cryopreservation and thaw process for cryopreserved MSCs

The duration of cryopreservation for cryopreserved MSCs prior to use in experiments was not reported in nine studies (*Devaney et al., 2015*; *Salmenkari et al., 2019*; *Tan et al., 2019*; *Curley et al., 2017*; *Bárcia et al., 2017*; *Horie et al., 2020a*; *Lohan et al., 2018*; *Perlee et al., 2019*; *Horie et al., 2020b*), four studies cryopreserved the MSCs for at least 1 month (*Somal et al., 2017*; *Horiuchi et al., 2021*; *Bharti et al., 2020*; *Rogulska et al., 2019*), and two for up to 2 months (*Horie et al., 2021*; *Yea et al., 2020*). One study cryopreserved MSCs for 2–3 weeks (*Khan et al., 2019*), another between 1 and 4 weeks (*Gramlich et al., 2016*), and one study cryopreserved their MSCs for 9 days (*Cruz et al., 2015*).

Ten studies used 10% DMSO (dimethyl sulfoxide) as part of their cryopreservation solution (*Cruz et al., 2015*; *Devaney et al., 2015*; *Salmenkari et al., 2019*; *Somal et al., 2017*; *Yea et al., 2020*; *Bárcia et al., 2017*; *Bharti et al., 2020*; *Khan et al., 2019*; *Lohan et al., 2018*; *Rogulska et al., 2019*), three studies used CryoStor Cell Preservation Media (Sigma-Aldrich) (*Gramlich et al., 2016*; *Horie et al., 2021*; *Horie et al., 2020a*), one study used MSC Freezing media (Biological Industries) (*Tan et al., 2019*), one study used 5% DMSO (*Horiuchi et al., 2021*), and three studies did not report the solution used for cryopreservation (*Curley et al., 2017*; *Perlee et al., 2019*; *Horie et al., 2020b*). Five studies did not report on their method of cryopreservation (*Devaney et al., 2015*; *Salmenkari et al., 2019*; *Horie et al., 2021*; *Horie et al., 2020a*; *Horie et al., 2020b*), three studies employed a controlled-rate freezer to achieve cryopreservation (*Tan et al., 2019*; *Curley et al., 2017*; *Bárcia et al., 2017*), while eight studies used liquid nitrogen at –80°C to –196°C (*Cruz et al., 2015*; *Gramlich et al., 2016*; *Somal et al., 2017*; *Yea et al., 2020*; *Bharti et al., 2020*; *Lohan et al., 2018*; *Perlee et al., 2019*; *Rogulska et al., 2019*) for storage, and two studies gradually cryopreserved the MSCs with decremental temperature over 24 hr, followed by storage at –150 °C (*Horiuchi et al., 2021*; *Khan et al., 2019*).

Eight studies did not report their thawing protocol (*Cruz et al., 2015*; *Devaney et al., 2015*; *Somal et al., 2017*; *Horie et al., 2021*; *Bárcia et al., 2017*; *Bharti et al., 2020*; *Horie et al., 2020a*;

**Table 2.** MSC characteristics of included studies.

| First author (Year) | Species and tissue source | Compatibility with animal | ISCT criteria met | Route of administration | Vehicle | Timing of MSCs post-disease induction | Fresh MSCs — Cryopreserved at any point? | Duration of culture | Cryopreserved MSCs — Method | Duration | Time from Thaw to Experiment |
|---|---|---|---|---|---|---|---|---|---|---|---|
| Bárcia et al., 2017 | Human Umbilical Cord | Xenogenic | Yes | 1) Intra-articular 2) Intra-muscular | PBS | 1) 7 days 2) 5 hr | No | >5 days | Controlled Rate Freezer | NR | Immediately |
| Cruz et al., 2015 | Human and Murine Bone Marrow | Syngenic and Xenogenic | Yes | Intravenous | PBS | 14 days | Yes | NR | –80°C for 48 hr then liquid nitrogen | 9 days | 15 min |
| Curley et al., 2017 | Human Umbilical Cord and Bone Marrow | Xenogenic | Yes | Intravenous | PBS | NR | No | 4 days | Controlled Rate Freezer | NR | Day of administration |
| Devaney et al., 2015 | Human Bone Marrow | Xenogenic | Yes | Intravenous | PBS | 0.5 hr | Yes | NR | NR | NR | 30 min |
| Gramlich et al., 2016 | Human | Xenogenic | Yes | Intra-ocular | PBS | 2 hr | Yes | >7 days | Controlled Rate Freezer | 7–30 days | <1 hr |
| Lohan et al., 2018 | Rat Bone Marrow | Allogenic | NR | Intravenous | PBS | 1 and 7 days prior | Yes | NR | –80°C for 24 hr then liquid nitrogen | NR | Immediately |
| Salmenkari et al., 2019 | Human Bone Marrow | Xenogenic | NR | Intravenous | 0.9% NaCl +3.6% HAS | 3 and 5 days | Yes | NR | NR | NR | NR |
| Somal et al., 2017 | Gravid caprine AF (amniotic fluid), AS (amniotic sac), WJ (Wharton's jelly), CB (cord blood) | Xenogenic | NR | Subcutaneously | PBS | 7, 14, 21, 28 days | Yes | NR | –80°C overnight then liquid nitrogen | Atleast 1 month | NR |
| Bharti et al., 2020 | Dog Bone Marrow | Xenogenic | NR | Surgically placed over wound | Polypropylene mesh | NR | Yes | NR | –80°C overnight then liquid nitrogen | 1 month | NR |
| Horie et al., 2020a | Human Bone Marrow | Xenogenic | NR | Intravenous | PBS | 6 hr | Yes | NR | NR | NR | NR |
| Horie et al., 2020a | Human Bone Marrow and Umbilical Cord | Xenogenic | NR | Intra-tracheal | PBS | 30 min | Yes | NR | NR | NR | Immediately |
| Khan et al., 2019 | Dog Adipose Tissue | Allogenic | NR | Intravenous | Hartmann's Solution | Immediately | Yes | NR | 4 °C for 1 hr, –20 °C for 2 hr, –80 °C for 24 hr, then –150 °C | 2–3 weeks | Immediately |
| Rogulska et al., 2019 | Human Adipose Tissue | Xenogenic | NR | Implantation into wound | 3D gel | Immediately | Yes | NR | –80°C the liquid nitrogen | 1 month | NR |
| Tan et al., 2019 | Human Bone Marrow | Xenogenic | Yes | Intravenous | 5% Human Albumin in PlasmaLyte | 6 hr | No | >24 hr | Controlled Rate Freezer | NR | Immediately |
| Perlee et al., 2019 | Human Adipose Tissue | Xenogenic | Yes | Intravenous | Ringer's Lactate | 1 or 6 hr | No | 24 hr | Liquid nitrogen | Until required | Day of administration |
| Yea et al., 2020 | Human Umbilical Cord | Xenogenic | NR | Intratendinous | PBS | Immediately | No | NR | –80°C then –196 °C Liquid Nitrogen | Up to 1 month | Immediately |
| Horiuchi et al., 2021 | Rat Synovial Fluid | Allogenic | NR | Intraarticular | PBS | Every week from 2 to 8 weeks | Yes | 7 days | –80 °C overnight, and then at –150 °C | 16 months | Immediately |
| Horie et al., 2021 | Human Umbilical Cord | Xenogenic | NR | Intravenous | PBS | 15 min | No | NR | NR | Up to 2 months | Immediately |

*Perlee et al., 2019*), one study employed a cell-thawing device called the ThawStar (AsteroBio, USA) (*Horiuchi et al., 2021*) and the remaining nine studies used a 37 °C hot water bath to thaw the cryo-preserved MSCs (*Gramlich et al., 2016*; *Salmenkari et al., 2019*; *Tan et al., 2019*; *Curley et al., 2017*; *Yea et al., 2020*; *Khan et al., 2019*; *Lohan et al., 2018*; *Rogulska et al., 2019*; *Horie et al., 2020b*). Two studies thawed MSCs on the day of administration for their experiments (*Curley et al., 2017*; *Perlee et al., 2019*), while nine studies reported thawing MSCs either immediately or within 1 hr of use in experimentation (*Cruz et al., 2015*; *Devaney et al., 2015*; *Gramlich et al., 2016*; *Tan et al., 2019*; *Yea et al., 2020*; *Bárcia et al., 2017*; *Khan et al., 2019*; *Lohan et al., 2018*; *Horie et al., 2020b*). Seven studies did not report time from thaw to use in experimentation (*Salmenkari et al., 2019*; *Somal et al., 2017*; *Horiuchi et al., 2021*; *Horie et al., 2021*; *Bharti et al., 2020*; *Horie et al., 2020a*; *Rogulska et al., 2019*). Nine studies suspended thawed MSCs in phosphate buffered saline (PBS, vehicle for experiments) (*Cruz et al., 2015*; *Devaney et al., 2015*; *Tan et al., 2019*; *Curley et al., 2017*; *Horiuchi et al., 2021*; *Bárcia et al., 2017*; *Khan et al., 2019*; *Lohan et al., 2018*; *Horie et al., 2020b*), while one study re-suspended them in ringer's lactate supplemented with 3% Dimethyl sulfoxide (DMSO) (*Perlee et al., 2019*), one used MSCs suspended in 0.9% NaCl +3.6% HSA (Human Serum Albumin) (*Salmenkari et al., 2019*), one used PBS with 5% HSA (*Tan et al., 2019*), and six studies did not report their resuspension solution (*Horie et al., 2021*; *Yea et al., 2020*; *Bharti et al., 2020*; *Horie et al., 2020a*; *Lohan et al., 2018*; *Rogulska et al., 2019*).

## Description of cryopreservation and culture process for freshly cultured MSCs

Freshly cultured MSCs were not cryopreserved at any point after harvest from source in 13 studies (range of total culture time: 4–28 days) (*Cruz et al., 2015*; *Devaney et al., 2015*; *Gramlich et al., 2016*; *Salmenkari et al., 2019*; *Somal et al., 2017*; *Horie et al., 2021*; *Yea et al., 2020*; *Bharti et al., 2020*; *Khan et al., 2019*; *Horie et al., 2020a*; *Lohan et al., 2018*; *Rogulska et al., 2019*; *Horie et al., 2020b*). In five studies, the MSCs were cryopreserved and then culture-expanded for more than 24 hr prior to use in experimentation (*Tan et al., 2019*; *Curley et al., 2017*; *Horiuchi et al., 2021*; *Bárcia et al., 2017*; *Perlee et al., 2019*).

Further details related to MSC culture, including medium, passage, concentration, and route of administration can be found in *Table 2*.

## Risk of bias

Of the 18 included studies, none of them met low-risk of bias criteria for all 10 domains and all studies demonstrated unclear risk of bias due to lack or reporting in atleast two domains. Ten studies did not have any features that would confer a high-risk of bias in the one of the 10 domains (*Cruz et al., 2015*; *Devaney et al., 2015*; *Tan et al., 2019*; *Curley et al., 2017*; *Horiuchi et al., 2021*; *Horie et al., 2021*; *Yea et al., 2020*; *Bharti et al., 2020*; *Khan et al., 2019*; *Horie et al., 2020a*). Five studies demonstrated high-risk of bias in one domain (*Devaney et al., 2015*; *Salmenkari et al., 2019*; *Somal et al., 2017*; *Perlee et al., 2019*; *Rogulska et al., 2019*), and the remaining three studies demonstrated high-risk of bias in two or more domains (*Gramlich et al., 2016*; *Bárcia et al., 2017*; *Lohan et al., 2018*). The complete reporting of the risk of bias domains is presented in *Table 3*.

## Primary and secondary outcomes

Across the 18 included studies, a total of 325 experiments and 133 distinct outcome measures were reported on our primary and secondary outcomes and are summarized below. Data extraction of outcomes from included studies yielded significant amounts of data given the extensive and varied inflammatory disease models and their specific outcomes. A description of all primary in vivo pre-clinical efficacy and secondary in vitro potency outcomes are reported in *Table 4* and 6, respectively. The studies included in our systematic review varied with respect to disease type, MSC source, MSC processing, route of administration, dose, outcome measures, and timing of outcome measurement. Due to this high degree of heterogeneity, meta-analyses were not feasible for the primary and secondary outcome measures. However, similar pre-clinical animal inflammatory models that reported similar outcomes are reported in *Table 5* for reference.

**Table 3.** Risk of Bias assessments for the included in vivo studies using SYRCLE Tool.

| Author (year) | Selection Bias | | | Performance Bias | | Detection Bias | | Attrition Bias | Reporting Bias | Other Bias |
|---|---|---|---|---|---|---|---|---|---|---|
| | Adequate randomization | Baseline charactersics given | Evidence of adequate concealment of groups | Evidence of random housing of animals | Evidence of caregivers blinded to intervention | Evidence of random selection for assessment | Evidence of assessor blinded | Explanation of missing animal data | Free of selective reporting based on methods/results | Free of other high bias risk |
| Bárcia et al., 2017 | Unclear | Yes (Low Risk) | Unclear | Yes (Low Risk) | No (High Risk) | Unclear | No (High Risk) | Yes (Low Risk) | Yes (Low Risk) | No (High Risk) |
| Bharti et al., 2020 | Unclear | Unclear | Unclear | Yes (Low Risk) | Unclear | Unclear | Unclear | Unclear | Yes (Low Risk) | Yes (Low Risk) |
| Cruz et al., 2015 | Unclear | Yes (Low Risk) | Unclear | Yes (Low Risk) | Unclear | Unclear | Yes (Low Risk) | Unclear | Yes (Low Risk) | Yes (Low Risk) |
| Curley et al., 2017 | Unclear | Yes (Low Risk) | Unclear | Unclear | Unclear | Unclear | Yes (Low Risk) | Unclear | Yes (Low Risk) | Yes (Low Risk) |
| Devaney et al., 2015 | Unclear | Yes (Low Risk) | Unclear | Unclear | Unclear | Unclear | No (High Risk) | Yes (Low Risk) | Yes (Low Risk) | Yes (Low Risk) |
| Gramlich et al., 2016 | No (High Risk) | Yes (Low Risk) | Unclear | Unclear | Unclear | Unclear | Yes (Low Risk) | Unclear | Yes (Low Risk) | No (High Risk) |
| Horie et al., 2020a | Unclear | Unclear | Unclear | Unclear | Unclear | Unclear | Yes (Low Risk) | Unclear | Yes (Low Risk) | Yes (Low Risk) |
| Horie et al., 2020a | Unclear | Unclear | Unclear | Unclear | Unclear | Unclear | Unclear | Yes (Low Risk) | Yes (Low Risk) | Yes (Low Risk) |
| Khan et al., 2019 | Unclear | Yes (Low Risk) | Unclear | Unclear | Yes (Low Risk) | Unclear | Yes (Low Risk) | Yes (Low Risk) | Yes (Low Risk) | Yes (Low Risk) |
| Lohan et al., 2018 | No (High Risk) | Unclear | Unclear | Unclear | Unclear | Unclear | Unclear | Unclear | No (High Risk) | Yes (Low Risk) |
| Perlee et al., 2019 | No (High Risk) | Unclear | Unclear | Yes (Low Risk) | Unclear | Unclear | Yes (Low Risk) | Unclear | Yes (Low Risk) | Yes (Low Risk) |
| Rogulska et al., 2019 | Unclear | Yes (Low Risk) | Unclear | Yes (Low Risk) | Unclear | Unclear | Yes (Low Risk) | Unclear | Yes (Low Risk) | No (High Risk) |
| Salmenkari et al., 2019 | No (High Risk) | Yes (Low Risk) | Unclear | Yes (Low Risk) | Unclear | Unclear | Yes (Low Risk) | Yes (Low Risk) | Yes (Low Risk) | Yes (Low Risk) |
| Somal et al., 2017 | No (High Risk) | Unclear | Unclear | Yes (Low Risk) | Unclear | Unclear | Unclear | Unclear | Yes (Low Risk) | Yes (Low Risk) |
| Tan et al., 2019 | Yes (Low Risk) | Yes (Low Risk) | Yes (Low Risk) | Unclear | Yes (Low Risk) | Unclear | Yes (Low Risk) | Yes (Low Risk) | Yes (Low Risk) | Yes (Low Risk) |
| Yea et al., 2020 | Unclear | Yes (Low Risk) | Unclear | Yes (Low Risk) | Unclear | Unclear | Unclear | Unclear | Yes (Low Risk) | Yes (Low Risk) |
| Horiuchi et al., 2021 | Unclear | Yes (Low Risk) | Unclear | Yes (Low Risk) | Unclear | Unclear | Unclear | Unclear | Yes (Low Risk) | Yes (Low Risk) |
| Horie et al., 2021 | Unclear | Yes (Low Risk) | Yes (Low Risk) | Yes (Low Risk) | Unclear | Unclear | Yes (Low Risk) | Unclear | Yes (Low Risk) | Yes (Low Risk) |

**Table 4.** All in vivo outcomes where freshly cultured vs. cryopreserved MSCs have been compared directly are reported.

| Study | Animal Model | Outcome | Number (n) | Type and Source of MSCs | Duration of Culture Post-Thaw (hr) | Concentration of MSCs | Pre-Treatment of MSCs | Negative Control (NC) | Positive Control (PC) | p-value for Fresh MSCs vs. control | p-value for Frozen MSCs vs. control | Fresh or Frozen MSC more effective? | p-value for Fresh vs. Frozen comparison |
|---|---|---|---|---|---|---|---|---|---|---|---|---|---|
| **Acute Lung Injury and Sepsis** | | | | | | | | | | | | | |
| *Devaney et al., 2015* | Acute lung injury induced by *E. coli* pneumonia in rats | Arterial oxygenation | 10 | Human Bone Marrow | 0 | 1×10^7 hMSCs/kg | N/A | N/A | PBS | <0.05 | <0.05 | ↔ | NS |
| | | Lung compliance | 10 | Human Bone Marrow | 0 | 1×10^7 hMSCs/kg | N/A | N/A | PBS | <0.05 | <0.05 | ↔ | NS |
| | | BAL protein | 10 | Human Bone Marrow | 0 | 1×10^7 hMSCs/kg | N/A | N/A | PBS | <0.05 | <0.05 | ↔ | NS |
| | | BAL neutrophils | 10 | Human Bone Marrow | 0 | 1×10^7 hMSCs/kg | N/A | N/A | PBS | <0.05 | <0.05 | ↔ | NS |
| | | BAL *E. coli* bacterial load | 10 | Human Bone Marrow | 0 | 1×10^7 hMSCs/kg | N/A | N/A | PBS | <0.05 | <0.05 | ↔ | NS |
| | | BAL IL-6 | 10 | Human Bone Marrow | 0 | 1×10^7 hMSCs/kg | N/A | N/A | PBS | <0.05 | <0.05 | ↔ | NS |
| | | BAL IL-10 | 10 | Human Bone Marrow | 0 | 1×10^7 hMSCs/kg | N/A | N/A | PBS | <0.05 | <0.05 | ↔ | NS |
| *Cruz et al., 2015* | Allergic Airways Inflammation induced by Aspergillus hyphal extract (AHE) exposure in mice. | Large Airway Resistance | 10 (Fresh) and 7 (Frozen) | Human Bone Marrow | 0 | 1 × 10^6 viable MSC cells | Frozen MSCs washed 3 times prior to use | Naïve (PBS model) | AHE+PBS, Human Lung Fibroblasts | <0.05 | <0.05 | ↔ | NS |
| | | Large Airway Resistance | 6 | Murine Bone Marrow | 0 | 1 × 10^6 viable MSC cells | Frozen MSCs washed 3 times prior to use | Naïve (PBS model) | AHE+PBS, Human Lung Fibroblasts | <0.05 | <0.05 | ↔ | NS |
| | | Overall Tissue Resistance | 10 (Fresh) and 7 (Frozen) | Human Bone Marrow | 0 | 1 × 10^6 viable MSC cells | Frozen MSCs washed 3 times prior to use | Naïve (PBS model) | AHE+PBS, Human Lung Fibroblasts | <0.05 | <0.05 | ↔ | NS |
| | | Overall Tissue Resistance | 6 | Murine Bone Marrow | 0 | 1 × 10^6 viable MSC cells | Frozen MSCs washed 3 times prior to use | Naïve (PBS model) | AHE+PBS, Human Lung Fibroblasts | <0.05 | <0.05 | ↔ | NS |
| | | Lung Elastance | 10 (Fresh) and 7 (Frozen) | Human Bone Marrow | 0 | 1 × 10^6 viable MSC cells | Frozen MSCs washed 3 times prior to use | Naïve (PBS model) | AHE+PBS, Human Lung Fibroblasts | <0.05 | <0.05 | ↔ | NS |
| | | Lung Elastance | 6 | Murine Bone Marrow | 0 | 1 × 10^6 viable MSC cells | Frozen MSCs washed 3 times prior to use | Naïve (PBS model) | AHE+PBS, Human Lung Fibroblasts | <0.05 | <0.05 | ↔ | NS |
| | | Inflammation Score | 10 (Fresh) and 7 (Frozen) | Human Bone Marrow | 0 | 1 × 10^6 viable MSC cells | Frozen MSCs washed 3 times prior to use | Naïve (PBS model) | AHE+PBS, Human Lung Fibroblasts | <0.05 | <0.05 | ↔ | NS |
| | | Inflammation Score | 6 | Murine Bone Marrow | 0 | 1 × 10^6 viable MSC cells | Frozen MSCs washed 3 times prior to use | Naïve (PBS model) | AHE+PBS, Human Lung Fibroblasts | <0.05 | <0.05 | ↔ | NS |
| | | BALF Total Cell Number | 10 (Fresh) and 7 (Frozen) | Human Bone Marrow | 0 | 1 × 10^6 viable MSC cells | Frozen MSCs washed 3 times prior to use | Naïve (PBS model) | AHE+PBS, Human Lung Fibroblasts | <0.05 | <0.05 | ↔ | NS |
| | | BALF Total Cell Number | 6 | Murine Bone Marrow | 0 | 1 × 10^6 viable MSC cells | Frozen MSCs washed 3 times prior to use | Naïve (PBS model) | AHE+PBS, Human Lung Fibroblasts | <0.05 | <0.05 | ↔ | NS |
| | | BAL Neutrophils | 10 (Fresh) and 7 (Frozen) | Human Bone Marrow | 0 | 1 × 10^6 viable MSC cells | Frozen MSCs washed 3 times prior to use | Naïve (PBS model) | AHE+PBS, Human Lung Fibroblasts | <0.05 | <0.05 | ↔ | NS |

*Table 4 continued on next page*

| | | | | | | | | | | | |
|---|---|---|---|---|---|---|---|---|---|---|---|
| BAL Neutrophils | 6 | Murine Bone Marrow | 0 | 1 × 10^6 viable MSC cells | Frozen MSCs washed 3 times prior to use | Naïve (PBS model) | AHE +PBS, Human Lung Fibroblasts | <0.05 | <0.05 | ↕ | NS |
| BAL Eosinophils | 10 (Fresh) and 7 (Frozen) | Human Bone Marrow | 0 | 1 × 10^6 viable MSC cells | Frozen MSCs washed 3 times prior to use | Naïve (PBS model) | AHE +PBS, Human Lung Fibroblasts | <0.05 | <0.05 | ↕ | NS |
| BAL Eosinophils | 6 | Murine Bone Marrow | 0 | 1 × 10^6 viable MSC cells | Frozen MSCs washed 3 times prior to use | Naïve (PBS model) | AHE +PBS, Human Lung Fibroblasts | <0.05 | <0.05 | ↕ | NS |
| BAL Macrophages | 10 (Fresh) and 7 (Frozen) | Human Bone Marrow | 0 | 1 × 10^6 viable MSC cells | Frozen MSCs washed 3 times prior to use | Naïve (PBS model) | AHE +PBS, Human Lung Fibroblasts | <0.05 | <0.05 | ↕ | NS |
| BAL Macrophages | 6 | Murine Bone Marrow | 0 | 1 × 10^6 viable MSC cells | Frozen MSCs washed 3 times prior to use | Naïve (PBS model) | AHE +PBS, Human Lung Fibroblasts | <0.05 | <0.05 | ↕ | NS |
| BAL Lymphocytes | 10 (Fresh) and 7 (Frozen) | Human Bone Marrow | 0 | 1 × 10^6 viable MSC cells | Frozen MSCs washed 3 times prior to use | Naïve (PBS model) | AHE +PBS, Human Lung Fibroblasts | <0.05 | <0.05 | ↕ | NS |
| BAL Lymphocytes | 6 | Murine Bone Marrow | 0 | 1 × 10^6 viable MSC cells | Frozen MSCs washed 3 times prior to use | Naïve (PBS model) | AHE +PBS, Human Lung Fibroblasts | <0.05 | <0.05 | Frozen better | <0.05 |
| BAL IL-1a | 10 (Fresh) and 7 (Frozen) | Human Bone Marrow | 0 | 1 × 10^6 viable MSC cells | Frozen MSCs washed 3 times prior to use | Naïve (PBS model) | AHE +PBS, Human Lung Fibroblasts | <0.05 | <0.05 | ↕ | NS |
| BAL IL-1a | 6 | Murine Bone Marrow | 0 | 1 × 10^6 viable MSC cells | Frozen MSCs washed 3 times prior to use | Naïve (PBS model) | AHE +PBS, Human Lung Fibroblasts | <0.05 | <0.05 | ↕ | NS |
| BAL IL-3 | 10 (Fresh) and 7 (Frozen) | Human Bone Marrow | 0 | 1 × 10^6 viable MSC cells | Frozen MSCs washed 3 times prior to use | Naïve (PBS model) | AHE +PBS, Human Lung Fibroblasts | <0.05 | <0.05 | ↕ | NS |
| BAL IL-3 | 6 | Murine Bone Marrow | 0 | 1 × 10^6 viable MSC cells | Frozen MSCs washed 3 times prior to use | Naïve (PBS model) | AHE +PBS, Human Lung Fibroblasts | <0.05 | <0.05 | ↕ | NS |
| BAL IL-4 | 10 (Fresh) and 7 (Frozen) | Human Bone Marrow | 0 | 1 × 10^6 viable MSC cells | Frozen MSCs washed 3 times prior to use | Naïve (PBS model) | AHE +PBS, Human Lung Fibroblasts | <0.05 | <0.05 | ↕ | NS |
| BAL IL-4 | 6 | Murine Bone Marrow | 0 | 1 × 10^6 viable MSC cells | Frozen MSCs washed 3 times prior to use | Naïve (PBS model) | AHE +PBS, Human Lung Fibroblasts | <0.05 | <0.05 | ↕ | NS |
| BAL IL-5 | 10 (Fresh) and 7 (Frozen) | Human Bone Marrow | 0 | 1 × 10^6 viable MSC cells | Frozen MSCs washed 3 times prior to use | Naïve (PBS model) | AHE +PBS, Human Lung Fibroblasts | <0.05 | <0.05 | ↕ | NS |
| BAL IL-5 | 6 | Murine Bone Marrow | 0 | 1 × 10^6 viable MSC cells | Frozen MSCs washed 3 times prior to use | Naïve (PBS model) | AHE +PBS, Human Lung Fibroblasts | <0.05 | <0.05 | ↕ | NS |
| BAL IL-6 | 10 (Fresh) and 7 (Frozen) | Human Bone Marrow | 0 | 1 × 10^6 viable MSC cells | Frozen MSCs washed 3 times prior to use | Naïve (PBS model) | AHE +PBS, Human Lung Fibroblasts | <0.05 | <0.05 | ↕ | NS |
| BAL IL-6 | 6 | Murine Bone Marrow | 0 | 1 × 10^6 viable MSC cells | Frozen MSCs washed 3 times prior to use | Naïve (PBS model) | AHE +PBS, Human Lung Fibroblasts | <0.05 | <0.05 | ↕ | NS |
| BAL IL-10 | 10 (Fresh) and 7 (Frozen) | Human Bone Marrow | 0 | 1 × 10^6 viable MSC cells | Frozen MSCs washed 3 times prior to use | Naïve (PBS model) | AHE +PBS, Human Lung Fibroblasts | <0.05 | <0.05 | ↕ | NS |
| BAL IL-10 | 6 | Murine Bone Marrow | 0 | 1 × 10^6 viable MSC cells | Frozen MSCs washed 3 times prior to use | Naïve (PBS model) | AHE +PBS, Human Lung Fibroblasts | <0.05 | <0.05 | ↕ | NS |
| BAL IL-12-p40 | 10 (Fresh) and 7 (Frozen) | Human Bone Marrow | 0 | 1 × 10^6 viable MSC cells | Frozen MSCs washed 3 times prior to use | Naïve (PBS model) | AHE +PBS, Human Lung Fibroblasts | <0.05 | <0.05 | ↕ | NS |

Table 4 continued

| Study | Measure | n | MSC Source | | Dose | Control 1 | Control 2 | Control 3 | p | p | Direction | Result |
|---|---|---|---|---|---|---|---|---|---|---|---|---|
| | BAL IL-12-p40 | 6 | Murine Bone Marrow | 0 | 1 × 10^6 viable MSC cells | Frozen MSCs washed 3 times prior to use | Naïve (PBS model) | AHE +PBS, Human Lung Fibroblasts | <0.05 | <0.05 | ↕ | NS |
| | BAL IL-13 | 10 (Fresh) and 7 (Frozen) | Human Bone Marrow | 0 | 1 × 10^6 viable MSC cells | Frozen MSCs washed 3 times prior to use | Naïve (PBS model) | AHE +PBS, Human Lung Fibroblasts | <0.05 | <0.05 | ↕ | NS |
| | BAL IL-13 | 6 | Murine Bone Marrow | 0 | 1 × 10^6 viable MSC cells | Frozen MSCs washed 3 times prior to use | Naïve (PBS model) | AHE +PBS, Human Lung Fibroblasts | <0.05 | <0.05 | ↕ | NS |
| | BAL IL-17 | 10 (Fresh) and 7 (Frozen) | Human Bone Marrow | 0 | 1 × 10^6 viable MSC cells | Frozen MSCs washed 3 times prior to use | Naïve (PBS model) | AHE +PBS, Human Lung Fibroblasts | <0.05 | <0.05 | Fresh better | <0.05 |
| | BAL IL-17 | 6 | Murine Bone Marrow | 0 | 1 × 10^6 viable MSC cells | Frozen MSCs washed 3 times prior to use | Naïve (PBS model) | AHE +PBS, Human Lung Fibroblasts | <0.05 | <0.05 | ↕ | NS |
| | BAL KC | 10 (Fresh) and 7 (Frozen) | Human Bone Marrow | 0 | 1 × 10^6 viable MSC cells | Frozen MSCs washed 3 times prior to use | Naïve (PBS model) | AHE +PBS, Human Lung Fibroblasts | <0.05 | <0.05 | Fresh better | <0.05 |
| | BAL KC | 6 | Murine Bone Marrow | 0 | 1 × 10^6 viable MSC cells | Frozen MSCs washed 3 times prior to use | Naïve (PBS model) | AHE +PBS, Human Lung Fibroblasts | <0.05 | <0.05 | Frozen better | <0.05 |
| | BAL RANTES | 10 (Fresh) and 7 (Frozen) | Human Bone Marrow | 0 | 1 × 10^6 viable MSC cells | Frozen MSCs washed 3 times prior to use | Naïve (PBS model) | AHE +PBS, Human Lung Fibroblasts | <0.05 | <0.05 | ↕ | NS |
| | BAL RANTES | 6 | Murine Bone Marrow | 0 | 1 × 10^6 viable MSC cells | Frozen MSCs washed 3 times prior to use | Naïve (PBS model) | AHE +PBS, Human Lung Fibroblasts | <0.05 | <0.05 | ↕ | NS |
| | IFN-y | 10 (Fresh) and 7 (Frozen) | Human Bone Marrow | 0 | 1 × 10^6 viable MSC cells | Frozen MSCs washed 3 times prior to use | Naïve (PBS model) | AHE +PBS, Human Lung Fibroblasts | <0.05 | <0.05 | ↕ | NS |
| | IFN-y | 6 | Murine Bone Marrow | 0 | 1 × 10^6 viable MSC cells | Frozen MSCs washed 3 times prior to use | Naïve (PBS model) | AHE +PBS, Human Lung Fibroblasts | <0.05 | <0.05 | ↕ | NS |
| Curley et al., 2017 | Arterial Oxygenation (FiO2=0.3) | 8–10 | Human Umbilical Cord (Frozen) and Bone marrow (Fresh) MSCs | NR | 1×10^7 MSCs/kg | N/A | Sham model +PBS | E. coli/+PBS | <0.05 | <0.05 | ↕ | NS |
| Acute respiratory distress syndrome by intratracheal installation of E. coli in rats. | Arterial Oxygenation (FiO2=1) | 8–10 | Human Umbilical Cord (Frozen) and Bone marrow (Fresh) MSCs | NR | 1×10^7 MSCs/kg | N/A | Sham model +PBS | E. coli/+PBS | <0.05 | <0.05 | ↕ | NS |
| | Lung Compliance | 8–10 | Human Umbilical Cord (Frozen) and Bone marrow (Fresh) MSCs | NR | 1×10^7 MSCs/kg | N/A | Sham model +PBS | E. coli/+PBS | <0.05 | <0.05 | ↕ | NS |
| | Wet:Dry Lung Ratio | 8–10 | Human Umbilical Cord (Frozen) and Bone marrow (Fresh) MSCs | NR | 1×10^7 MSCs/kg | N/A | Sham model +PBS | E. coli/+PBS | <0.05 | <0.05 | ↕ | NS |
| | BAL Neutrophils | 8–10 | Human Umbilical Cord (Frozen) and Bone marrow (Fresh) MSCs | NR | 1×10^7 MSCs/kg | N/A | Sham model +PBS | E. coli/+PBS | <0.05 | <0.05 | ↕ | NS |

Table 4 continued on next page

*Table 4 continued*

| Study | Model | Outcome | N | Source of MSCs | | Dose | Cell processing | Control 1 | Control 2 | P-value 1 | P-value 2 | | |
|---|---|---|---|---|---|---|---|---|---|---|---|---|---|
| **Bárcia et al., 2017** | 1) Chronic adjuvant-induced arthritis (AIA) model 2) Hindlimb ischemia model in mice | BAL Bacteria | 8–10 | Human Umbilical Cord (Frozen) and Bone marrow (Fresh) MSCs | NR | $1\times10^7$ MSCs/kg | N/A | Sham model +PBS | E. coli +PBS | <0.05 | <0.05 | ↕ | NS |
| | | Arthritis Index | 6 | Human Umbilical Cord MSCs | 0 | $1.7\times10^6$ MSCs | Fresh MSCs were cryopreserved and then cultured for up to 5 days | Sham model +PBS | N/A | P<0.0001 | P<0.0001 | ↕ | NS |
| | | Left Paw Volume | 6 | Human Umbilical Cord MSCs | 0 | $1.7\times10^6$ MSCs | Fresh MSCs were cryopreserved and then cultured for up to 5 days | Sham model +PBS | N/A | P<0.0001 | P<0.0001 | ↕ | NS |
| | | Right Paw Volume | 6 | Human Umbilical Cord MSCs | 0 | $1.7\times10^6$ MSCs | Fresh MSCs were cryopreserved and then cultured for up to 5 days | Sham model +PBS | N/A | P<0.0001 | P<0.0001 | ↕ | NS |
| | | Weight | 6 | Human Umbilical Cord MSCs | 0 | $1.7\times10^6$ MSCs | Fresh MSCs were cryopreserved and then cultured for up to 5 days | Sham model +PBS | N/A | P<0.0001 | P<0.0001 | ↕ | NS |
| | | Blood Flow Ratio in Hindlimb D0 | 12 | Human Umbilical Cord MSCs | 0 | $2\times10^5$ MSCs | Fresh MSCs were cryopreserved and then cultured for up to 5 days | N/A | PBS | NS | NS | ↕ | NS |
| | | Blood Flow Ratio in Hindlimb D7 | 12 | Human Umbilical Cord MSCs | 0 | $2\times10^5$ MSCs | Fresh MSCs were cryopreserved and then cultured for up to 5 days | N/A | PBS | P=0.008 | P=0.019 | ↕ | NS |
| | | Blood Flow Ratio in Hindlimb D14 | 12 | Human Umbilical Cord MSCs | 0 | $2\times10^5$ MSCs | Fresh MSCs were cryopreserved and then cultured for up to 5 days | N/A | PBS | P=0.012 | P=0.031 | ↕ | NS |
| | | Blood Flow Ratio in Hindlimb D21 | 12 | Human Umbilical Cord MSCs | 0 | $2\times10^5$ MSCs | Fresh MSCs were cryopreserved and then cultured for up to 5 days | N/A | PBS | P=0.004 | P=0.002 | ↕ | NS |
| **Salmenkari et al., 2019** | Acute phase and Regenerative Phase of Colitis model in mice | Macroscopic Score | 9 | Human Bone Marrow | NR | $0.5\times10^6$ MSCs | N/A | Sham model with PBS | Colitis +Vehicle | PC: NS | PC: NS | ↕ | NS |
| | | Colon Weight (% change) | 9 | Human Bone Marrow | NR | $0.5\times10^6$ MSCs | N/A | Sham model with PBS | Colitis +Vehicle | PC: NS NC = P=0.001 | PC: NS NC: P=0.001 | ↕ | NS |
| | | Colon Length | 9 | Human Bone Marrow | NR | $0.5\times10^6$ MSCs | N/A | Sham model with PBS | Colitis +Vehicle | PC: NS NC = P=0.018 | PC: NS NC: P=0.014 | ↕ | NS |
| | | Histopathology Score | 9 | Human Bone Marrow | NR | $0.5\times10^6$ MSCs | N/A | Sham model with PBS | Colitis +Vehicle | PC: NS NC = P=0.004 | PC: NS NC: P=0.001 | ↕ | NS |
| | | Regeneration | 9 | Human Bone Marrow | NR | $0.5\times10^6$ MSCs | N/A | Sham model with PBS | Colitis +Vehicle | PC: NS | PC: NS | ↕ | NS |
| | | IL-1b in colon tissue homogenates | 9 | Human Bone Marrow | NR | $0.5\times10^6$ MSCs | N/A | Sham model with PBS | Colitis +Vehicle | PC: NS | PC: NS | ↕ | NS |
| | | TNFa in colon tissue homogenates | 9 | Human Bone Marrow | NR | $0.5\times10^6$ MSCs | N/A | Sham model with PBS | Colitis +Vehicle | PC: NS | PC: NS | ↕ | NS |
| | | IL-1b mRNA in colon | 9 | Human Bone Marrow | NR | $0.5\times10^6$ MSCs | N/A | Sham model with PBS | Colitis +Vehicle | PC: NS | PC: NS | ↕ | NS |
| | | Corticosterone in colon tissue homogenates | 9 | Human Bone Marrow | NR | $0.5\times10^6$ MSCs | N/A | Sham model with PBS | Colitis +Vehicle | PC: NS | PC: NS | ↕ | NS |

*Table 4 continued on next page*

*Table 4 continued*

| Study / Model | Parameter | n | Source | | Dose | | Control 1 | Control 2 | | | | |
|---|---|---|---|---|---|---|---|---|---|---|---|---|
| | Tissue ACE levels | 9 | Human Bone Marrow | NR | 0.5 × 10^6 MSCs | N/A | Sham model with PBS | Colitis +Vehicle | PC: NS | PC: P<0.05 | ↕ | NS |
| | Atgr1a mRNA expression | 9 | Human Bone Marrow | NR | 0.5 × 10^6 MSCs | N/A | Sham model with PBS | Colitis +Vehicle | PC: NS | PC: NS | ↕ | NS |
| | ACE shedding | 9 | Human Bone Marrow | NR | 0.5 × 10^6 MSCs | N/A | Sham model with PBS | Colitis +Vehicle | PC: NS | PC: P<0.001 | ↕ | NS |
| **Somal et al., 2017** — Wound Healing of surgical dorsal limb wound in rats | Wound Area D0 | 3 | Caprine Amniotic Fluid | NR | 1 × 10^6 MSC cells | N/A | N/A | PBS | NS | NS | ↕ | NS |
| | Wound Area D7 | 3 | Caprine Amniotic Fluid | NR | 1 × 10^6 MSC cells | N/A | N/A | PBS | P<0.05 | P<0.05 | ↕ | NS |
| | Wound Area D14 | 3 | Caprine Amniotic Fluid | NR | 1 × 10^6 MSC cells | N/A | N/A | PBS | NS | NS | ↕ | NS |
| | Wound Area D21 | 3 | Caprine Amniotic Fluid | NR | 1 × 10^6 MSC cells | N/A | N/A | PBS | NS | NS | ↕ | NS |
| | Wound Area D28 | 3 | Caprine Amniotic Fluid | NR | 1 × 10^6 MSC cells | N/A | N/A | PBS | NS | NS | ↕ | NS |
| | % Wound Contraction D7 | 3 | Caprine Amniotic Fluid | NR | 1 × 10^6 MSC cells | N/A | N/A | PBS | P<0.05 | NS | ↕ | NS |
| | % Wound Contraction D14 | 3 | Caprine Amniotic Fluid | NR | 1 × 10^6 MSC cells | N/A | N/A | PBS | NS | NS | ↕ | NS |
| | % Wound Contraction D21 | 3 | Caprine Amniotic Fluid | NR | 1 × 10^6 MSC cells | N/A | N/A | PBS | NS | NS | ↕ | NS |
| | % Wound Contraction D28 | 3 | Caprine Amniotic Fluid | NR | 1 × 10^6 MSC cells | N/A | N/A | PBS | NS | NS | ↕ | NS |
| | Epithelization | 3 | Caprine Amniotic Fluid | NR | 1 × 10^6 MSC cells | N/A | N/A | PBS | P<0.05 | P<0.05 | ↕ | NS |
| | Neovascularization | 3 | Caprine Amniotic Fluid | NR | 1 × 10^6 MSC cells | N/A | N/A | PBS | P<0.05 | P<0.05 | ↕ | NS |
| | Collagen Thickness | 3 | Caprine Amniotic Fluid | NR | 1 × 10^6 MSC cells | N/A | N/A | PBS | P<0.05 | P<0.05 | ↕ | NS |
| | Collagen Density | 3 | Caprine Amniotic Fluid | NR | 1 × 10^6 MSC cells | N/A | N/A | PBS | P<0.05 | P<0.05 | ↕ | NS |
| | Wound Area D0 | 3 | Caprine Amniotic Sac | NR | 1 × 10^6 MSC cells | N/A | N/A | PBS | NS | NS | ↕ | NS |
| | Wound Area D7 | 3 | Caprine Amniotic Sac | NR | 1 × 10^6 MSC cells | N/A | N/A | PBS | NS | NS | ↕ | NS |
| | Wound Area D14 | 3 | Caprine Amniotic Sac | NR | 1 × 10^6 MSC cells | N/A | N/A | PBS | NS | NS | ↕ | NS |
| | Wound Area D21 | 3 | Caprine Amniotic Sac | NR | 1 × 10^6 MSC cells | N/A | N/A | PBS | NS | NS | ↕ | NS |
| | Wound Area D28 | 3 | Caprine Amniotic Sac | NR | 1 × 10^6 MSC cells | N/A | N/A | PBS | NS | NS | ↕ | NS |

*Table 4 continued*

| | n | Source | | Dose | | | | | | | |
|---|---|---|---|---|---|---|---|---|---|---|---|
| % Wound Contraction D7 | 3 | Caprine Amniotic Sac | NR | 1 × 10^6 MSC cells | N/A | N/A | PBS | NS | NS | ↕ | NS |
| % Wound Contraction D14 | 3 | Caprine Amniotic Sac | NR | 1 × 10^6 MSC cells | N/A | N/A | PBS | NS | NS | ↕ | NS |
| % Wound Contraction D21 | 3 | Caprine Amniotic Sac | NR | 1 × 10^6 MSC cells | N/A | N/A | PBS | NS | NS | ↕ | NS |
| % Wound Contraction D28 | 3 | Caprine Amniotic Sac | NR | 1 × 10^6 MSC cells | N/A | N/A | PBS | NS | NS | ↕ | NS |
| Epithelization | 3 | Caprine Amniotic Sac | NR | 1 × 10^6 MSC cells | N/A | N/A | PBS | $P<0.05$ | $P<0.05$ | ↕ | NS |
| Neovascularization | 3 | Caprine Amniotic Sac | NR | 1 × 10^6 MSC cells | N/A | N/A | PBS | $P<0.05$ | $P<0.05$ | ↕ | NS |
| Collagen Thickness | 3 | Caprine Amniotic Sac | NR | 1 × 10^6 MSC cells | N/A | N/A | PBS | NS | $P<0.05$ | ↕ | NS |
| Collagen Density | 3 | Caprine Amniotic Sac | NR | 1 × 10^6 MSC cells | N/A | N/A | PBS | $P<0.05$ | $P<0.05$ | ↕ | NS |
| Wound Area D0 | 3 | Caprine Wharton's Jelly | NR | 1 × 10^6 MSC cells | N/A | N/A | PBS | NS | NS | ↕ | NS |
| Wound Area D7 | 3 | Caprine Wharton's Jelly | NR | 1 × 10^6 MSC cells | N/A | N/A | PBS | $P<0.05$ | NS | ↕ | NS |
| Wound Area D14 | 3 | Caprine Wharton's Jelly | NR | 1 × 10^6 MSC cells | N/A | N/A | PBS | NS | NS | ↕ | NS |
| Wound Area D21 | 3 | Caprine Wharton's Jelly | NR | 1 × 10^6 MSC cells | N/A | N/A | PBS | NS | NS | ↕ | NS |
| Wound Area D28 | 3 | Caprine Wharton's Jelly | NR | 1 × 10^6 MSC cells | N/A | N/A | PBS | $P<0.05$ | NS | ↕ | NS |
| % Wound Contraction D7 | 3 | Caprine Wharton's Jelly | NR | 1 × 10^6 MSC cells | N/A | N/A | PBS | NS | NS | ↕ | NS |
| % Wound Contraction D14 | 3 | Caprine Wharton's Jelly | NR | 1 × 10^6 MSC cells | N/A | N/A | PBS | NS | NS | ↕ | NS |
| % Wound Contraction D21 | 3 | Caprine Wharton's Jelly | NR | 1 × 10^6 MSC cells | N/A | N/A | PBS | NS | NS | ↕ | NS |
| % Wound Contraction D28 | 3 | Caprine Wharton's Jelly | NR | 1 × 10^6 MSC cells | N/A | N/A | PBS | NS | NS | ↕ | NS |
| Epithelization | 3 | Caprine Wharton's Jelly | NR | 1 × 10^6 MSC cells | N/A | N/A | PBS | $P<0.05$ | $P<0.05$ | ↕ | NS |
| Neovascularization | 3 | Caprine Wharton's Jelly | NR | 1 × 10^6 MSC cells | N/A | N/A | PBS | $P<0.05$ | $P<0.05$ | ↕ | NS |
| Collagen Thickness | 3 | Caprine Wharton's Jelly | NR | 1 × 10^6 MSC cells | N/A | N/A | PBS | $P<0.05$ | $P<0.05$ | ↕ | NS |
| Collagen Density | 3 | Caprine Wharton's Jelly | NR | 1 × 10^6 MSC cells | N/A | N/A | PBS | $P<0.05$ | $P<0.05$ | ↕ | NS |

*Table 4 continued on next page*

*Table 4 continued*

| Study | Model | Measurement | n | Cell source | Time to injection | Dose | Comparison detail | Comparator | Control | P-value (a) | P-value (b) | Direction | Result |
|---|---|---|---|---|---|---|---|---|---|---|---|---|---|
| | | Wound Area D0 | 3 | Caprine Cord Blood | NR | 1 × 10^6 MSC cells | N/A | N/A | PBS | NS | NS | ↕ | NS |
| | | Wound Area D7 | 3 | Caprine Cord Blood | NR | 1 × 10^6 MSC cells | N/A | N/A | PBS | P<0.05 | NS | ↕ | NS |
| | | Wound Area D14 | 3 | Caprine Cord Blood | NR | 1 × 10^6 MSC cells | N/A | N/A | PBS | NS | NS | ↕ | NS |
| | | Wound Area D21 | 3 | Caprine Cord Blood | NR | 1 × 10^6 MSC cells | N/A | N/A | PBS | NS | NS | ↕ | NS |
| | | Wound Area D28 | 3 | Caprine Cord Blood | NR | 1 × 10^6 MSC cells | N/A | N/A | PBS | NS | NS | ↕ | NS |
| | | % Wound Contraction D7 | 3 | Caprine Cord Blood | NR | 1 × 10^6 MSC cells | N/A | N/A | PBS | P<0.05 | NS | ↕ | NS |
| | | % Wound Contraction D14 | 3 | Caprine Cord Blood | NR | 1 × 10^6 MSC cells | N/A | N/A | PBS | NS | NS | ↕ | NS |
| | | % Wound Contraction D21 | 3 | Caprine Cord Blood | NR | 1 × 10^6 MSC cells | N/A | N/A | PBS | NS | NS | ↕ | NS |
| | | % Wound Contraction D28 | 3 | Caprine Cord Blood | NR | 1 × 10^6 MSC cells | N/A | N/A | PBS | NS | NS | ↕ | NS |
| | | Epithelization | 3 | Caprine Cord Blood | NR | 1 × 10^6 MSC cells | N/A | N/A | PBS | P<0.05 | P<0.05 | ↕ | NS |
| | | Neovascularization | 3 | Caprine Cord Blood | NR | 1 × 10^6 MSC cells | N/A | N/A | PBS | P<0.05 | P<0.05 | ↕ | NS |
| | | Collagen Thickness | 3 | Caprine Cord Blood | NR | 1 × 10^6 MSC cells | N/A | N/A | PBS | NS | NS | ↕ | NS |
| | | Collagen Density | 3 | Caprine Cord Blood | NR | 1 × 10^6 MSC cells | N/A | N/A | PBS | P<0.05 | NS | Frozen better | P<0.05 |
| *Lohan et al., 2018* | Corneal Transplantation in rats | Opacity Score, measured from day 5 post-implantation to day 30 | Fresh = 13, Frozen = 10 | Rat Bone Marrow | 0 | 1×10^6 MSC | Frozen MSCs pre-treated with allogenic splenocytes, and co-intervention with MMF. No MMF for Fresh MSCs. | N/A | Transplantation +No treatment | NS | | ↕ | NR |
| | | Neovascularization Score, measured from day 5 post-implantation to day 30 | Fresh = 13, Frozen = 10 | Rat Bone Marrow | 0 | 1×10^6 MSC | Frozen MSCs pre-treated with allogenic splenocytes, and co-intervention with MMF. No MMF for Fresh MSCs. | N/A | Transplantation +No treatment | P<0.001 | NS | ↕ | NR |
| *Gramlich et al., 2016* | Retinal ischemia/reperfusion model in mice | Retinal ganglion cells/mm^2 | Fresh = 10, Frozen = 8 | Human MSCs | <1 hr | 3×10^4 MSC | N/A | Sham model | PBS | P=0.019 | P=0.024 | ↕ | NS |
| *Perlee et al., 2019* | Pneumosepsis Caused by Klebsiella pneumoniae | Lung Bacterial Load at 16 hours | 8 | Human Adipose Tissue | 0 | 1×10^6 ASCs | MSCs infused at 1 or 6 hours after infection. | N/A | PBS | NS | P<0.001 | ↕ | NS |
| | | Lung Bacterial Load at 48 hours | 8 | Human Adipose Tissue | 0 | 1×10^6 ASCs | MSCs infused at 1 or 6 hours after infection. | N/A | PBS | P<0.0001 | P<0.001 | ↕ | NS |
| | | Blood Bacterial Load at 16 hours | 8 | Human Adipose Tissue | 0 | 1×10^6 ASCs | MSCs infused at 1 or 6 hours after infection. | N/A | PBS | NS | NS | ↕ | NS |
| | | Blood Bacterial Load at 48 hours | 8 | Human Adipose Tissue | 0 | 1×10^6 ASCs | MSCs infused at 1 or 6 hours after infection. | N/A | PBS | P<0.001 | P<0.001 | ↕ | NS |

*Table 4 continued on next page*

*Table 4 continued*

| | n | Source | | Dose | Timing | | Comparator | | | | |
|---|---|---|---|---|---|---|---|---|---|---|---|
| Liver Bacterial Load at 16 hours | 8 | Human Adipose Tissue | 0 | 1×10^6 ASCs | MSCs infused at 1 or 6 hours after infection. | N/A | PBS | NS | NS | ↕ | NS |
| Liver Bacterial Load at 48 hours | 8 | Human Adipose Tissue | 0 | 1×10^6 ASCs | MSCs infused at 1 or 6 hours after infection. | N/A | PBS | $P<0.0001$ | $P<0.001$ | ↕ | NS |
| Spleen Bacterial Load at 16 hours | 8 | Human Adipose Tissue | 0 | 1×10^6 ASCs | MSCs infused at 1 or 6 hours after infection. | N/A | PBS | NS | NS | ↕ | NS |
| Spleen Bacterial Load at 48 hours | 8 | Human Adipose Tissue | 0 | 1×10^6 ASCs | MSCs infused at 1 or 6 hours after infection. | N/A | PBS | $P<0.001$ | $P<0.01$ | ↕ | NS |
| Lung TNFa at 16 hours | 8 | Human Adipose Tissue | 0 | 1×10^6 ASCs | MSCs infused at 1 or 6 hours after infection. | N/A | PBS | $P<0.0001$ | $P<0.05$ | ↕ | NS |
| Lung TNFa at 48 hours | 8 | Human Adipose Tissue | 0 | 1×10^6 ASCs | MSCs infused at 1 or 6 hours after infection. | N/A | PBS | $P<0.001$ | $P<0.05$ | ↕ | NS |
| Lung IL-1b at 16 hours | 8 | Human Adipose Tissue | 0 | 1×10^6 ASCs | MSCs infused at 1 or 6 hours after infection. | N/A | PBS | $P<0.05$ | $P<0.01$ | ↕ | NS |
| Lung IL-1b at 48 hours | 8 | Human Adipose Tissue | 0 | 1×10^6 ASCs | MSCs infused at 1 or 6 hours after infection. | N/A | PBS | $P<0.001$ | $P<0.05$ | ↕ | NS |
| Lung IL-6 at 16 hours | 8 | Human Adipose Tissue | 0 | 1×10^6 ASCs | MSCs infused at 1 or 6 hours after infection. | N/A | PBS | $P<0.05$ | $P<0.01$ | ↕ | NS |
| Lung IL-6 at 48 hours | 8 | Human Adipose Tissue | 0 | 1×10^6 ASCs | MSCs infused at 1 or 6 hours after infection. | N/A | PBS | $P<0.01$ | NS | ↕ | NS |
| MIP-2 at 16 hours | 8 | Human Adipose Tissue | 0 | 1×10^6 ASCs | MSCs infused at 1 or 6 hours after infection. | N/A | PBS | $P<0.05$ | $P<0.01$ | ↕ | NS |
| MIP-2 at 48 hours | 8 | Human Adipose Tissue | 0 | 1×10^6 ASCs | MSCs infused at 1 or 6 hours after infection. | N/A | PBS | $P<0.001$ | $P<0.05$ | ↕ | NS |
| **Horie et al., 2020a**  *E. coli-induced lung injury.*  Arterial Oxygenation | 8 | Human Umbilical Cord | 0 | 1×10^7 MSCs/kg | Isolated CD362+MSCs for use | N/A | PBS | $P<0.05$ | $P<0.05$ | ↕ | NS |
| Lung Wet:Dry Ratio | 8 | Human Umbilical Cord | 0 | 1×10^7 MSCs/kg | Isolated CD362+MSCs for use | N/A | PBS | NS | NS | ↕ | NS |
| Lung Compliance | 8 | Human Umbilical Cord | 0 | 1×10^7 MSCs/kg | Isolated CD362+MSCs for use | N/A | PBS | $P<0.05$ | NS | ↕ | NS |
| BAL E. coli Counts | 8 | Human Umbilical Cord | 0 | 1×10^7 MSCs/kg | Isolated CD362+MSCs for use | N/A | PBS | $P<0.05$ | $P<0.05$ | ↕ | NS |
| BAL WCC levels | 8 | Human Umbilical Cord | 0 | 1×10^7 MSCs/kg | Isolated CD362+MSCs for use | N/A | PBS | $P<0.05$ | $P<0.05$ | ↕ | NS |
| BAL Neutrophils | 8 | Human Umbilical Cord | 0 | 1×10^7 MSCs/kg | Isolated CD362+MSCs for use | N/A | PBS | $P<0.05$ | $P<0.05$ | ↕ | NS |
| BAL IL-1b | 8 | Human Umbilical Cord | 0 | 1×10^7 MSCs/kg | Isolated CD362+MSCs for use | N/A | PBS | $P<0.05$ | $P<0.05$ | ↕ | NS |
| BAL CINC-1 | 8 | Human Umbilical Cord | 0 | 1×10^7 MSCs/kg | Isolated CD362+MSCs for use | N/A | PBS | NS | NS | ↕ | NS |

*Table 4 continued on next page*

*Table 4 continued*

| Study | Model | Outcome | n | Cell source | Passage | Dose | Cell type | Control | Vehicle | P-value | P-value | Direction | Significance |
|---|---|---|---|---|---|---|---|---|---|---|---|---|---|
| | | BAL IL-6 | 8 | Human Umbilical Cord | 0 | | N/A | N/A | PBS | P<0.05 | P<0.05 | ↕ | NS |
| Horie et al., 2020a | Ventilator-induced Lung Injury | Arterial Oxygenation | Fresh, n=7–8; Cryopreserved, n=5–6 | Human Bone Marrow | NR | 1×10^7 MSCs/kg | Pre-activated MSCs (fresh and frozen were also used) | Sham model | PBS | P<0.001 | P<0.001 | ↕ | NS |
| | | Lung Compliance | Fresh, n=7–8; Cryopreserved, n=5–6 | Human Bone Marrow | NR | 1×10^7 MSCs/kg | Pre-activated MSCs (fresh and frozen were also used) | Sham model | PBS | NS | NS | ↕ | NS |
| | | Lung Wet:Dry Ratio | Fresh, n=7–8; Cryopreserved, n=5–6 | Human Bone Marrow | NR | 1×10^7 MSCs/kg | Pre-activated MSCs (fresh and frozen were also used) | Sham model | PBS | P<0.05 | P<0.05 | ↕ | NS |
| | | BAL Protein | Fresh, n=7–8; Cryopreserved, n=5–6 | Human Bone Marrow | NR | 1×10^7 MSCs/kg | Pre-activated MSCs (fresh and frozen were also used) | Sham model | PBS | NS | NS | ↕ | NS |
| | | Percentage of Alveolar Airspace | Fresh, n=8; Cryopreserved, n=6 | Human Bone Marrow | NR | 1×10^7 MSCs/kg | Pre-activated MSCs (fresh and frozen were also used) | Sham model | PBS | P<0.001 | P<0.001 | ↕ | NS |
| | | BAL Neutrophils | Fresh, n=6–8; Cryopreserved, n=5–6 | Human Bone Marrow | NR | 1×10^7 MSCs/kg | Pre-activated MSCs (fresh and frozen were also used) | Sham model | PBS | P<0.05 | P<0.01 | ↕ | NS |
| | | BAL CINC-1 | Fresh, n=6–8; Cryopreserved, n=5–6 | Human Bone Marrow | NR | 1×10^7 MSCs/kg | Pre-activated MSCs (fresh and frozen were also used) | Sham model | PBS | P<0.05 | P<0.05 | ↕ | NS |
| | | BAL IL-6 | Fresh, n=6–8; Cryopreserved, n=5–6 | Human Bone Marrow | NR | 1×10^7 MSCs/kg | Pre-activated MSCs (fresh and frozen were also used) | Sham model | PBS | P<0.05 | P<0.001 | ↕ | NS |
| | | BAL IL-10 | Fresh, n=6–8; Cryopreserved, n=5–6 | Human Bone Marrow | NR | 1×10^7 MSCs/kg | Pre-activated MSCs (fresh and frozen were also used) | Sham model | PBS | NS | NS | ↕ | NS |
| | | BAL KGF | Fresh, n=6–8; Cryopreserved, n=5–6 | Human Bone Marrow | NR | 1×10^7 MSCs/kg | Pre-activated MSCs (fresh and frozen were also used) | Sham model | PBS | NS | NS | ↕ | NS |
| | | BAL PGE2 | Fresh, n=6–8; Cryopreserved, n=5–6 | Human Bone Marrow | NR | 1×10^7 MSCs/kg | Pre-activated MSCs (fresh and frozen were also used) | Sham model | PBS | NS | NS | ↕ | NS |
| Tan et al., 2019 | Polymicrobial sepsis induced by cecal-ligation-and-puncture (CLP) | %CD11b+/E. coli+cells in Peritoneal Fluid | Fresh, n=12; Cryopreserved, n=11 | Human Bone Marrow | 0 | 2.5×10^5 MSC cells | N/A | N/A | PBS | P<0.0001 | P<0.0001 | ↕ | NS |
| | | Peritoneal CFU # | Fresh, n=12; Cryopreserved, n=11 | Human Bone Marrow | 0 | 2.5×10^5 MSC cells | N/A | N/A | PBS | NS | NS | ↕ | NS |
| | | Plasma Lactate | Fresh, n=12; Cryopreserved, n=11 | Human Bone Marrow | 0 | 2.5×10^5 MSC cells | N/A | N/A | PBS | P<0.05 | P<0.05 | ↕ | NS |
| | | Plasma CCL5 | Fresh, n=12; Cryopreserved, n=11 | Human Bone Marrow | 0 | 2.5×10^5 MSC cells | N/A | N/A | PBS | NS | P<0.01 | ↕ | NS |
| | | Plasma JE | Fresh, n=12; Cryopreserved, n=11 | Human Bone Marrow | 0 | 2.5×10^5 MSC cells | N/A | N/A | PBS | NS | NS | ↕ | NS |
| | | Plasma KC | Fresh, n=12; Cryopreserved, n=11 | Human Bone Marrow | 0 | 2.5×10^5 MSC cells | N/A | N/A | PBS | P<0.05 | NS | ↕ | NS |
| | | Plasma LIX | Fresh, n=12; Cryopreserved, n=11 | Human Bone Marrow | 0 | 2.5×10^5 MSC cells | N/A | N/A | PBS | NS | NS | ↕ | NS |
| | | Plasma IL-10 | Fresh, n=12; Cryopreserved, n=11 | Human Bone Marrow | 0 | 2.5×10^5 MSC cells | N/A | N/A | PBS | NS | NS | ↕ | NS |
| | | Plasma IL-1b | Fresh, n=12; Cryopreserved, n=11 | Human Bone Marrow | 0 | 2.5×10^5 MSC cells | N/A | N/A | PBS | NS | NS | ↕ | NS |

*Table 4 continued on next page*

*Table 4 continued*

| Study | Model | Outcome | Day | Tissue source | Time | Cell dose | Intervention | | Control | P-value | P-value | Direction | NS |
|---|---|---|---|---|---|---|---|---|---|---|---|---|---|
| **Bharti et al., 2020** | Wound healing model with 2×2 cm^2 full-thickness excision skin wound in guinea pigs | Percent wound contraction D7 | 5 | Dog Bone Marrow | NR | 1×10^6 MSC cells | MSCs attached to polypropylene mesh of 2×2 cm2 size | N/A | Antibiotic only, Mesh only and MSCs only as control groups | NS | NS | ↕ | NS |
| | | Percent wound contraction D14 | 5 | Dog Bone Marrow | NR | 1×10^6 MSC cells | MSCs attached to polypropylene mesh of 2×2 cm2 size | N/A | Antibiotic only, Mesh only and MSCs only as control groups | P<0.05 | P<0.05 | ↕ | NS |
| | | Percent wound contraction D21 | 5 | Dog Bone Marrow | NR | 1×10^6 MSC cells | MSCs attached to polypropylene mesh of 2×2 cm2 size | N/A | Antibiotic only, Mesh only and MSCs only as control groups | P<0.05 | P<0.05 | ↕ | NS |
| | | Percent wound contraction D28 | 5 | Dog Bone Marrow | NR | 1×10^6 MSC cells | MSCs attached to polypropylene mesh of 2×2 cm2 size | N/A | Antibiotic only, Mesh only and MSCs only as control groups | P<0.05 | P<0.05 | ↕ | NS |
| | | Epithelialization | 5 | Dog Bone Marrow | NR | 1×10^6 MSC cells | MSCs attached to polypropylene mesh of 2×2 cm2 size | N/A | Antibiotic only, Mesh only and MSCs only as control groups | P<0.05 | P<0.05 | ↕ | NS |
| | | Neovascularization | 5 | Dog Bone Marrow | NR | 1×10^6 MSC cells | MSCs attached to polypropylene mesh of 2×2 cm2 size | N/A | Antibiotic only, Mesh only and MSCs only as control groups | P<0.05 | P<0.05 | ↕ | NS |
| | | Collagen Density | 5 | Dog Bone Marrow | NR | 1×10^6 MSC cells | MSCs attached to polypropylene mesh of 2×2 cm2 size | N/A | Antibiotic only, Mesh only and MSCs only as control groups | P<0.05 | P<0.05 | ↕ | NS |
| | | Collagen Thickness | 5 | Dog Bone Marrow | NR | 1×10^6 MSC cells | MSCs attached to polypropylene mesh of 2×2 cm2 size | N/A | Antibiotic only, Mesh only and MSCs only as control groups | P<0.05 | P<0.05 | ↕ | NS |
| **Rogulska et al., 2019** | Wound Healing of Full-thickness excisional skin wounds in mice | Percent Wound Closure D3 | 14 | Human Adipose Tissue | 24 hours | 0.25–0.3×10^6 cells in 50 µl | MSCs placed on 3D gel containing PPP, 0.2 M sucrose, 1% DMSO | N/A | Spontaneous healing, and 3D gel containing PPP, 0.2 M sucrose, 1% DMSO alone | P<0.05 | P<0.05 | ↕ | NS |
| | | Percent Wound Closure D7 | 14 | Human Adipose Tissue | 24 hours | 0.25–0.3×10^6 cells in 50 µl | MSCs placed on 3D gel containing PPP, 0.2 M sucrose, 1% DMSO | N/A | Spontaneous healing, and 3D gel containing PPP, 0.2 M sucrose, 1% DMSO alone | P<0.05 | P<0.05 | ↕ | NS |
| | | Percent Wound Closure D14 | 14 | Human Adipose Tissue | 24 hours | 0.25–0.3×10^6 cells in 50 µl | MSCs placed on 3D gel containing PPP, 0.2 M sucrose, 1% DMSO | N/A | Spontaneous healing, and 3D gel containing PPP, 0.2 M sucrose, 1% DMSO alone | P<0.05 | P<0.05 | ↕ | NS |
| | | Percent Wound Closure D28 | 14 | Human Adipose Tissue | 24 hours | 0.25–0.3×10^6 cells in 50 µl | MSCs placed on 3D gel containing PPP, 0.2 M sucrose, 1% DMSO | N/A | Spontaneous healing, and 3D gel containing PPP, 0.2 M sucrose, 1% DMSO alone | P<0.05 | P<0.05 | ↕ | NS |
| **Khan et al., 2019** | Acute Spinal Cord Injury in dogs | Motor activity of hind limbs assessed by using the canine Basso Beattie Bresnahan (cBBB) score at Week 1 | 4 | Dog Adipose Tissue | 0 | 1×10^7 MSC cells | Lentivirus Mediated HO-1 Gene Insertion into Ad-MSCs. | N/A | Fresh MSCs expressing GFP only. | NS | NS | ↕ | NS |
| | | cBBB score at Week 2 | 4 | Dog Adipose Tissue | 0 | 1×10^7 MSC cells | Lentivirus Mediated HO-1 Gene Insertion into Ad-MSCs. | N/A | Fresh MSCs expressing GFP only. | NS | NS | ↕ | NS |
| | | cBBB score at Week 3 | 4 | Dog Adipose Tissue | 0 | 1×10^7 MSC cells | Lentivirus Mediated HO-1 Gene Insertion into Ad-MSCs. | N/A | Fresh MSCs expressing GFP only. | NS | NS | ↕ | NS |
| | | cBBB score at Week 4 | 4 | Dog Adipose Tissue | 0 | 1×10^7 MSC cells | Lentivirus Mediated HO-1 Gene Insertion into Ad-MSCs. | N/A | Fresh MSCs expressing GFP only. | P<0.05 | NS | ↕ | NS |

*Table 4 continued*

| Study | Outcome measure | n | Cell source | Passage | Cell dose | Intervention | Comparator | Control | | | | |
|---|---|---|---|---|---|---|---|---|---|---|---|---|
| | % age of gross lesion area | 4 | Dog Adipose Tissue | 0 | 1×10^6 MSC cells | Lentivirus Mediated HO-1 Gene Insertion into Ad-MSCs. | N/A | Fresh MSCs expressing GFP only. | NS | NS | | NS |
| | Fibrotic areas relative to normal | 4 | Dog Adipose Tissue | 0 | 1×10^7 MSC cells | Lentivirus Mediated HO-1 Gene Insertion into Ad-MSCs. | Normal (no SCI) | Fresh MSCs expressing GFP only. | P<0.05 | NS | ↕ | NS |
| | Myelinated areas relative to normal | 4 | Dog Adipose Tissue | 0 | 1×10^7 MSC cells | Lentivirus Mediated HO-1 Gene Insertion into Ad-MSCs. | Normal (no SCI) | Fresh MSCs expressing GFP only. | P<0.05 | NS | ↕ | NS |
| **Yea et al., 2020** Wound healing in rats | Total macroscopic score at 2 weeks | 4 | Human Umbilical Cord | NR | 1×10^6 MSC cells | N/A | Cryoprotectant and PBS | Fresh-MSCs | P=0.001 | P=0.04 | ↕ | NS |
| | Total macroscopic score at 4 weeks | 4 | Human Umbilical Cord | NR | 1×10^6 MSC cells | N/A | Cryoprotectant and PBS | Fresh-MSCs | P=0.001 | P<0.05 | ↕ | NS |
| | Total degeneration score at 2 weeks | 4 | Human Umbilical Cord | NR | 1×10^6 MSC cells | N/A | Cryoprotectant and PBS | Fresh-MSCs | P<0.001 | P<0.001 | ↕ | NS |
| | Total degeneration score at 4 weeks | 4 | Human Umbilical Cord | NR | 1×10^6 MSC cells | N/A | Cryoprotectant and PBS | Fresh-MSCs | P<0.05 | P<0.05 | ↕ | NS |
| | Fibre structure at 2 weeks | 4 | Human Umbilical Cord | NR | 1×10^6 MSC cells | N/A | Cryoprotectant and PBS | Fresh-MSCs | NS | NS | ↕ | NS |
| | Fibre structure at 4 weeks | 4 | Human Umbilical Cord | NR | 1×10^6 MSC cells | N/A | Cryoprotectant and PBS | Fresh-MSCs | P<0.05 | P<0.05 | ↕ | NS |
| | Fibre arrangement at 2 weeks | 4 | Human Umbilical Cord | NR | 1×10^6 MSC cells | N/A | Cryoprotectant and PBS | Fresh-MSCs | NS | NS | ↕ | NS |
| | Fibre arrangement at 4 weeks | 4 | Human Umbilical Cord | NR | 1×10^6 MSC cells | N/A | Cryoprotectant and PBS | Fresh-MSCs | P<0.05 | P<0.05 | ↕ | NS |
| | Rounding of nuclei at 2 weeks | 4 | Human Umbilical Cord | NR | 1×10^6 MSC cells | N/A | Cryoprotectant and PBS | Fresh-MSCs | NS | NS | ↕ | NS |
| | Rounding of nuclei at 4 weeks | 4 | Human Umbilical Cord | NR | 1×10^6 MSC cells | N/A | Cryoprotectant and PBS | Fresh-MSCs | P<0.05 | P<0.05 | ↕ | NS |
| | Variations in cellularity at 2 weeks | 4 | Human Umbilical Cord | NR | 1×10^6 MSC cells | N/A | Cryoprotectant and PBS | Fresh-MSCs | NS | NS | ↕ | NS |
| | Variations in cellularity at 4 weeks | 4 | Human Umbilical Cord | NR | 1×10^6 MSC cells | N/A | Cryoprotectant and PBS | Fresh-MSCs | P<0.05 | P<0.05 | ↕ | NS |
| | Decreased stainability at 2 weeks | 4 | Human Umbilical Cord | NR | 1×10^6 MSC cells | N/A | Cryoprotectant and PBS | Fresh-MSCs | NS | NS | ↕ | NS |
| | Decreased stainability at 4 weeks | 4 | Human Umbilical Cord | NR | 1×10^6 MSC cells | N/A | Cryoprotectant and PBS | Fresh-MSCs | P<0.05 | P<0.05 | ↕ | NS |
| | Hyalinization at 2 weeks | 4 | Human Umbilical Cord | NR | 1×10^6 MSC cells | N/A | Cryoprotectant and PBS | Fresh-MSCs | NS | NS | ↕ | NS |
| | Hyalinization at 4 weeks | 4 | Human Umbilical Cord | NR | 1×10^6 MSC cells | N/A | Cryoprotectant and PBS | Fresh-MSCs | P<0.05 | P<0.05 | ↕ | NS |

*Table 4 continued on next page*

*Table 4 continued*

| | | | | | | | | | | | |
|---|---|---|---|---|---|---|---|---|---|---|---|
| Inflammation at 2 weeks | 4 | Human Umbilical Cord | NR | 1×10^6 MSC cells | N/A | Cryoprotectant and PBS | Fresh-MSCs | NS | NS | ↕ | NS |
| Inflammation at 4 weeks | 4 | Human Umbilical Cord | NR | 1×10^6 MSC cells | N/A | Cryoprotectant and PBS | Fresh-MSCs | P<0.05 | P<0.05 | ↕ | NS |
| Fibroblast density at 2 weeks | 4 | Human Umbilical Cord | NR | 1×10^6 MSC cells | N/A | Cryoprotectant and PBS | Fresh-MSCs | NS | NS | ↕ | NS |
| Fibroblast density at 4 weeks | 4 | Human Umbilical Cord | NR | 1×10^6 MSC cells | N/A | Cryoprotectant and PBS | Fresh-MSCs | P<0.05 | P<0.05 | ↕ | NS |
| Nuclear aspect ratio at 2 weeks | 4 | Human Umbilical Cord | NR | 1×10^6 MSC cells | N/A | Cryoprotectant and PBS | Fresh-MSCs | NS | NS | ↕ | NS |
| Nuclear aspect ration at 4 weeks | 4 | Human Umbilical Cord | NR | 1×10^6 MSC cells | N/A | Cryoprotectant and PBS | Fresh-MSCs | P<0.05 | P<0.05 | ↕ | NS |
| Nuclear orientation at 2 weeks | 4 | Human Umbilical Cord | NR | 1×10^6 MSC cells | N/A | Cryoprotectant and PBS | Fresh-MSCs | P<0.05 | P<0.05 | ↕ | NS |
| Nuclear orientation at 4 weeks | 4 | Human Umbilical Cord | NR | 1×10^6 MSC cells | N/A | Cryoprotectant and PBS | Fresh-MSCs | P<0.05 | P<0.05 | ↕ | NS |
| Collagen organization at 2 weeks | 4 | Human Umbilical Cord | NR | 1×10^6 MSC cells | N/A | Cryoprotectant and PBS | Fresh-MSCs | P<0.05 | P<0.05 | ↕ | NS |
| Collagen organization at 4 weeks | 4 | Human Umbilical Cord | NR | 1×10^6 MSC cells | N/A | Cryoprotectant and PBS | Fresh-MSCs | P<0.05 | P<0.05 | ↕ | NS |
| Collagen fibre coherence at 2 weeks | 4 | Human Umbilical Cord | NR | 1×10^6 MSC cells | N/A | Cryoprotectant and PBS | Fresh-MSCs | NS | NS | ↕ | NS |
| Collagen fibre coherence at 4 weeks | 4 | Human Umbilical Cord | NR | 1×10^6 MSC cells | N/A | Cryoprotectant and PBS | Fresh-MSCs | P<0.05 | P<0.05 | ↕ | NS |
| GAG-rich area at 2 weeks | 4 | Human Umbilical Cord | NR | 1×10^6 MSC cells | N/A | Cryoprotectant and PBS | Fresh-MSCs | P<0.05 | P<0.05 | ↕ | NS |
| GAG-rich area at 4 weeks | 4 | Human Umbilical Cord | NR | 1×10^6 MSC cells | N/A | Cryoprotectant and PBS | Fresh-MSCs | P<0.05 | P<0.05 | ↕ | NS |
| Ultimate failure load at 2 weeks | 8 | Human Umbilical Cord | NR | 1×10^6 MSC cells | N/A | Cryoprotectant and PBS | Fresh-MSCs | P<0.05 | P<0.05 | ↕ | NS |
| Ultimate failure load at 4 weeks | 8 | Human Umbilical Cord | NR | 1×10^6 MSC cells | N/A | Cryoprotectant and PBS | Fresh-MSCs | P<0.05 | P<0.05 | ↕ | NS |
| Tendon stiffness at 2 weeks | 8 | Human Umbilical Cord | NR | 1×10^6 MSC cells | N/A | Cryoprotectant and PBS | Fresh-MSCs | P<0.05 | P<0.05 | ↕ | NS |
| Tendon stiffness at 4 weeks | 8 | Human Umbilical Cord | NR | 1×10^6 MSC cells | N/A | Cryoprotectant and PBS | Fresh-MSCs | NS | NS | ↕ | NS |
| Ultimate stress at 2 weeks | 8 | Human Umbilical Cord | NR | 1×10^6 MSC cells | N/A | Cryoprotectant and PBS | Fresh-MSCs | P<0.05 | P<0.05 | ↕ | NS |
| Ultimate stress at 4 weeks | 8 | Human Umbilical Cord | NR | 1×10^6 MSC cells | N/A | Cryoprotectant and PBS | Fresh-MSCs | P<0.05 | P<0.05 | ↕ | NS |

*Table 4 continued*

| Study | Model | Measure | n | Cell source | Passage | Dose | | Vehicle | Group | | | | |
|---|---|---|---|---|---|---|---|---|---|---|---|---|---|
| | | Cross-sectional area at 2 weeks | 8 | Human Umbilical Cord | NR | 1×10^6 MSC cells | N/A | Cryoprotectant and PBS | Fresh-MSCs | P<0.05 | P<0.05 | ↕ | NS |
| | | Cross-sectional area at 4 weeks | 8 | Human Umbilical Cord | NR | 1×10^6 MSC cells | N/A | Cryoprotectant and PBS | Fresh-MSCs | P<0.05 | P<0.05 | ↕ | NS |
| | | Bioluminescence | 9 | Rat synovial MSCs | NR | 1×10^6 MSC cells | N/A | PBS | Fresh-MSCs | NR | NR | ↕ | NS |
| | | Tibia gross finding score | 9 | Rat synovial MSCs | NR | 1×10^6 MSC cells | N/A | PBS | Fresh-MSCs | P<0.05 | P<0.05 | ↕ | NS |
| | | Femur gross finding score | 9 | Rat synovial MSCs | NR | 1×10^6 MSC cells | N/A | PBS | Fresh-MSCs | P<0.05 | P<0.05 | ↕ | NS |
| | | Tibia OARSI score | 6 | Rat synovial MSCs | NR | 1×10^6 MSC cells | N/A | PBS | Fresh-MSCs | P<0.05 | P<0.05 | ↕ | NS |
| Horiuchi et al., 2021 | Osteoarthritis model in rats | Femur OARSI score | 6 | Rat synovial MSCs | NR | 1×10^6 MSC cells | N/A | PBS | Fresh-MSCs | NS | NS | ↕ | NS |
| Horie et al., 2021 | Ventilator-Induced Lung Injury (VILI) model in rats | Arterial oxygenation | 7 | Human Umbilical Cord MSCs | NR | 1 × 10^7 MSCs/kg | N/A | PBS | Fresh MSCs | P<0.001 | P<0.001 | ↕ | NS |
| | | Static Lung Compliance | 7 | Human Umbilical Cord MSCs | NR | 1 × 10^7 MSCs/kg | N/A | PBS | Fresh MSCs | P<0.01 | P<0.01 | ↕ | NS |
| | | Wet:Dry Ratio | 7 | Human Umbilical Cord MSCs | NR | 1 × 10^7 MSCs/kg | N/A | PBS | Fresh MSCs | P<0.05 | P<0.05 | ↕ | NS |
| | | BAL Protein | 7 | Human Umbilical Cord MSCs | NR | 1 × 10^7 MSCs/kg | N/A | PBS | Fresh MSCs | P<0.01 | P<0.01 | ↕ | NS |
| | | BAL Cell count | 7 | Human Umbilical Cord MSCs | NR | 1 × 10^7 MSCs/kg | N/A | PBS | Fresh MSCs | P<0.01 | P<0.01 | ↕ | NS |
| | | BAL Neutrophil count | 7 | Human Umbilical Cord MSCs | NR | 1 × 10^7 MSCs/kg | N/A | PBS | Fresh MSCs | P<0.05 | P<0.05 | ↕ | NS |
| | | BAL IL-6 level | 7 | Human Umbilical Cord MSCs | NR | 1 × 10^7 MSCs/kg | N/A | PBS | Fresh MSCs | NS | P<0.05 | Frozen better | P<0.05 |
| | | BAL IL-1 level | 7 | Human Umbilical Cord MSCs | NR | 1 × 10^7 MSCs/kg | N/A | PBS | Fresh MSCs | P<0.05 | P<0.05 | ↕ | NS |
| | | % Airspace | 4 | Human Umbilical Cord MSCs | NR | 1 × 10^7 MSCs/kg | N/A | PBS | Fresh MSCs | P<0.001 | P<0.001 | ↕ | NS |

↕ indicates no statistically significant difference of Freshly-cultured and Cryopreserved MSCs.

NS indicates Not Significant- statistical analysis from individual studies did not yield significant difference between Freshly-cultured and Cryopreserved MSCs. NR = Not reported.

If direct comparison of Freshly-cultured vs. Cryopreserved MSC was not presented in the same graph by a study, the results and discussion sections of that study were used to judge efficacy of Freshly-cultured vs. Cryopreserved MSCs for the table above.

## Primary outcomes

### In vivo pre-clinical efficacy outcomes

The 18 studies reported a total of 257 experiments and 101 distinct outcome measures related to our in vivo pre-clinical efficacy primary outcomes. Seventeen studies assessed composition of tissues (*Cruz et al., 2015*; *Devaney et al., 2015*; *Gramlich et al., 2016*; *Salmenkari et al., 2019*; *Somal et al., 2017*; *Tan et al., 2019*; *Curley et al., 2017*; *Horiuchi et al., 2021*; *Horie et al., 2021*; *Yea et al., 2020*; *Bárcia et al., 2017*; *Bharti et al., 2020*; *Khan et al., 2019*; *Horie et al., 2020a*; *Lohan et al., 2018*; *Perlee et al., 2019*; *Rogulska et al., 2019*), and 12 assessed organ dysfunction (*Cruz et al., 2015*; *Devaney et al., 2015*; *Gramlich et al., 2016*; *Salmenkari et al., 2019*; *Curley et al., 2017*; *Horiuchi et al., 2021*; *Horie et al., 2021*; *Yea et al., 2020*; *Bárcia et al., 2017*; *Khan et al., 2019*; *Horie et al., 2020a*; *Horie et al., 2020b*). Eleven of the 18 studies assessed protein expression and secretion (*Cruz et al., 2015*; *Devaney et al., 2015*; *Salmenkari et al., 2019*; *Tan et al., 2019*; *Curley et al., 2017*; *Horiuchi et al., 2021*; *Khan et al., 2019*; *Horie et al., 2020a*; *Lohan et al., 2018*; *Perlee et al., 2019*; *Horie et al., 2020b*) (*Table 2*).

Of the 257 experiments, six outcomes were significantly different at the 0.05 level or less, with two that favoured freshly cultured and four that favoured cryopreserved MSCs (*Table 4*).

### In vivo pre-clinical efficacy: function and composition of tissue

Seventeen studies reported organ dysfunction and/or composition of tissue outcomes and a total of 166 experiments were reported across the studies. Of the 116 experiments, only one reported a significant difference at the 0.05 level or less between the freshly cultured and cryopreserved MSC groups which favoured the cryopreserved group (*Figure 2*).

### In vivo pre-clinical efficacy: protein (cytokine) expression and secretion

Eleven studies reported protein expression and secretion outcomes, with total of 91 experiments reported across the studies. Five of the 91 experiments reported a statistically significant difference between freshly cultured and cryopreserved MSCs that were derived from one study (*Cruz et al., 2015*). Of the five experiments that demonstrated a significant difference at the 0.05 level or less, two favoured freshly cultured and three favoured cryopreserved MSCs (*Figure 3*).

## Secondary outcomes

### In vitro potency outcomes

Fifteen studies reported in vitro potency outcomes, including viability (*Cruz et al., 2015*; *Devaney et al., 2015*; *Gramlich et al., 2016*; *Somal et al., 2017*; *Tan et al., 2019*; *Curley et al., 2017*; *Horiuchi et al., 2021*; *Bárcia et al., 2017*; *Bharti et al., 2020*; *Khan et al., 2019*; *Horie et al., 2020a*; *Lohan et al., 2018*; *Perlee et al., 2019*; *Rogulska et al., 2019*; *Horie et al., 2020b*) with 68 experiments and 32 different outcome measures. All reported in vitro outcomes can be found in *Table 6*. Of the 68 experiments, 9 were significantly different at the 0.05 level or less, with 7 that favoured freshly cultured and 2 that favoured cryopreserved MSCs (*Figure 4*).

### In vitro potency: protein (cytokine) expression and secretion

A total of four studies (*Gramlich et al., 2016*; *Horiuchi et al., 2021*; *Bharti et al., 2020*; *Khan et al., 2019*) reported in vitro protein (cytokine) expression and secretion outcomes. Of the 33 experiments, five demonstrated a significant difference at the 0.05 level or less, with two favouring cryopreserved and three favouring freshly cultured MSCs (*Table 5*).

### In vitro potency: co-culture assays

Three studies reported in vitro co-culture assay outcomes (7 separate experiments) to assess the impact of MSCs on responder cell proliferation (*Gramlich et al., 2016*; *Tan et al., 2019*; *Bárcia et al., 2017*). All three studies used PBMCs (peripheral blood mononuclear cell) activated with CD3 and CD28 as the responder cells. The studies employed variable MSC:Responder cell ratios and duration of culture. All three studies found no significant difference in potency for cryopreserved as compared to freshly-cultured MSCs at varying concentrations of MSCs to responder cells (*Table 7*).

**Table 5.** Summary of similar in-vivo outcomes reported across studies.

| Outcome Measure | Study | Unit of Measurement | Number of samples (n) | Fresh MSC Mean | Fresh MSC Std Dev | Frozen MSC Mean | Frozen MSC Std Dev |
|---|---|---|---|---|---|---|---|
| | Curley et al., 2017 | mmHg | 8 to 10 | 217.77 | 77.93 | 242.75 | 84.14 |
| | Devaney et al., 2015 | mmHg | 10 | 265.5 | 67.86 | 247.64 | 68.232 |
| | Horie et al., 2020a | mmHg | 8 | 73.084 | 11.526 | 69.148 | 9.222 |
| Arterial Oxygenation0.128 | Horie et al., 2021 | kPa | 7 | 16.52 | 0.85 | 16.86 | 1.10 |
| | Curley et al., 2017 | mL/mmHg | 8 to 10 | 0.862 | 0.082 | 0.818 | 0.098 |
| | Devaney et al., 2015 | mL/mmHg | 12 | 0.82264 | 0.132 | 0.765 | 0.128 |
| | Horie et al., 2020a | mL/mmHg | 8 | 0.55939 | 0.089 | 0.451 | 0.531 |
| Lung Compliance | Horie et al., 2021 | mL/cmH2O | 7 | 0.363 | 0.06 | 0.358 | 0.08 |
| | Curley et al., 2017 | Ratio | 8 to 10 | 4.72779 | 0.188 | 4.77 | 0.157 |
| | Horie et al., 2020a | Ratio | 8 | 4.7643 | 0.074 | 4.94 | 0.294 |
| Wet:Dry Lung Ratio | Horie et al., 2021 | Ratio | 7 | 5.21 | 0.36 | 5.32 | 0.42 |
| | Devaney et al., 2015 | pg/ml | 12 | 348.93 | 207.5 | 363.22 | 142.5 |
| | Horie et al., 2020a | pg/ml | 8 | 224.67 | 119.86 | 181.51 | 126.72 |
| BAL IL-6 levels | Horie et al., 2021 | pg/ml | 7 | 252.39 | 61.64 | 207.76 | 53.66 |
| | Somal et al., 2017 | Percentage | 3 | 60.076 | 16.67 | 55.679 | 12.755 |
| | Bharti et al., 2020 | Percentage | 5 | 16.104 | 1.062 | 14.521 | 2.123 |
| % of Wound Contraction on D7 | Rogulska et al., 2019 | Percentage | 14 | 51.402 | 5.741 | 52.069 | 4.94 |
| | Somal et al., 2017 | Percentage | 3 | 96.374 | 0.85 | 89.937 | 5.103 |
| | Bharti et al., 2020 | Percentage | 5 | 67.363 | 1.69 | 71.537 | 2.123 |
| % of Wound Contraction on D14 | Rogulska et al., 2019 | Percentage | 14 | 99.065 | 2.8 | 99.866 | 2.804 |
| | Somal et al., 2017 | Percentage | 3 | 99.85 | 0.681 | 98.515 | 2.89 |
| % of Wound Contraction on D21 | Bharti et al., 2020 | Percentage | 5 | 84.141 | 1.93 | 89.457 | 1.769 |
| | Somal et al., 2017 | Percentage | 3 | 100.433 | | 100.288 | 0.681 |
| % of Wound Contraction on D28 | Bharti et al., 2020 | Percentage | 5 | 99.583 | 0.885 | 99.415 | 0.885 |

## Viability

Seventeen studies (*Cruz et al., 2015*; *Devaney et al., 2015*; *Gramlich et al., 2016*; *Somal et al., 2017*; *Tan et al., 2019*; *Curley et al., 2017*; *Horiuchi et al., 2021*; *Horie et al., 2021*; *Yea et al., 2020*; *Bárcia et al., 2017*; *Bharti et al., 2020*; *Khan et al., 2019*; *Horie et al., 2020a*; *Lohan et al., 2018*; *Perlee et al., 2019*; *Rogulska et al., 2019*; *Horie et al., 2020b*) reported post-thaw viability of cryopreserved MSCs, the range was from 60% to 98% across various time points since thawing. The viability of freshly cultured MSCs ranged from 91% to 99%, also assessed at various time points. Only seven studies reported on 25 viability experiments which compared viability directly between freshly cultured and cryopreserved MSCs (*Gramlich et al., 2016*; *Somal et al., 2017*; *Tan et al., 2019*; *Horiuchi et al., 2021*; *Horie et al., 2021*; *Yea et al., 2020*; *Bárcia et al., 2017*) Of the 25 experiments, 9 (36%) favoured freshly cultured MSCs (*Figure 5*).

## Discussion

Our study is the first comprehensive pre-clinical systematic review to examine the effect of cryopreservation on the in vivo efficacy and in vitro potency of MSCs in animal models of inflammation. Across the 18 included studies, our review found that 251 out of 257 (97.6%) of the in vivo pre-clinical efficacy outcomes demonstrated no statistically significant differences between cryopreserved and freshly cultured MSCs at a p value of<0.05. When evaluating the results of a large, heterogeneous group of studies with different outcome measures comparing freshly cultured versus cryopreserved MSCs for efficacy and potency, it is useful to compare the results to what one would expect to see

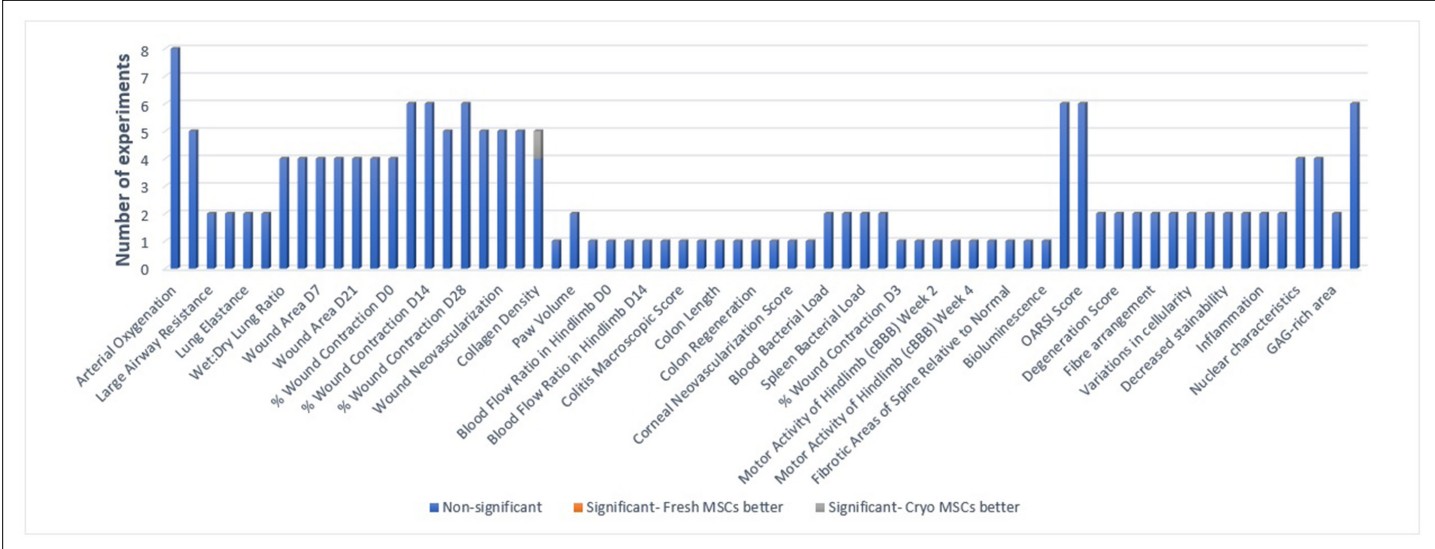

**Figure 2.** Primary in vivo outcomes. All the outcomes related to function and composition of tissues are presented below. Number of experiments represent the number of separate comparisons between freshly cultured and cryopreserved MSCs on surrogate measures of in vivo efficacy.

if (a) there were truly no difference or if (b) there truly were a difference. In the former case, where all differences would be due exclusively to Type I error, we would expect to see roughly 5% of the p-values as statistically significant. Furthermore, when a difference was statistically significant, we would expect it to be equally likely to favor freshly cultured versus cryopreserved or vice versa. In the latter case, where there truly is a difference, we would expect to see more than 5% of the p-values of all experiments as statistically significant and a strong concordance in the sense that most would favor the same group. We argue that our results for in vivo preclinical efficacy are consistent with pure Type 1 error (2.6% were statistically significant with roughly half favoring freshly cultured and half favoring cryopreserved MSCs). For in vitro potency, the results are somewhat less clear cut. We found 13% (95% Confidence Interval: 5–21%) were significantly different; 7 favored freshly cultured and 2 favored cryopreserved MSCs. Given that the confidence interval for the rate of statistical significance does not exclude 5% and that 2 of the 9 significant results favored cryopreserved MSCs, it does not represent strong evidence of a significant difference in in vitro potency. In terms of viability, the evidence

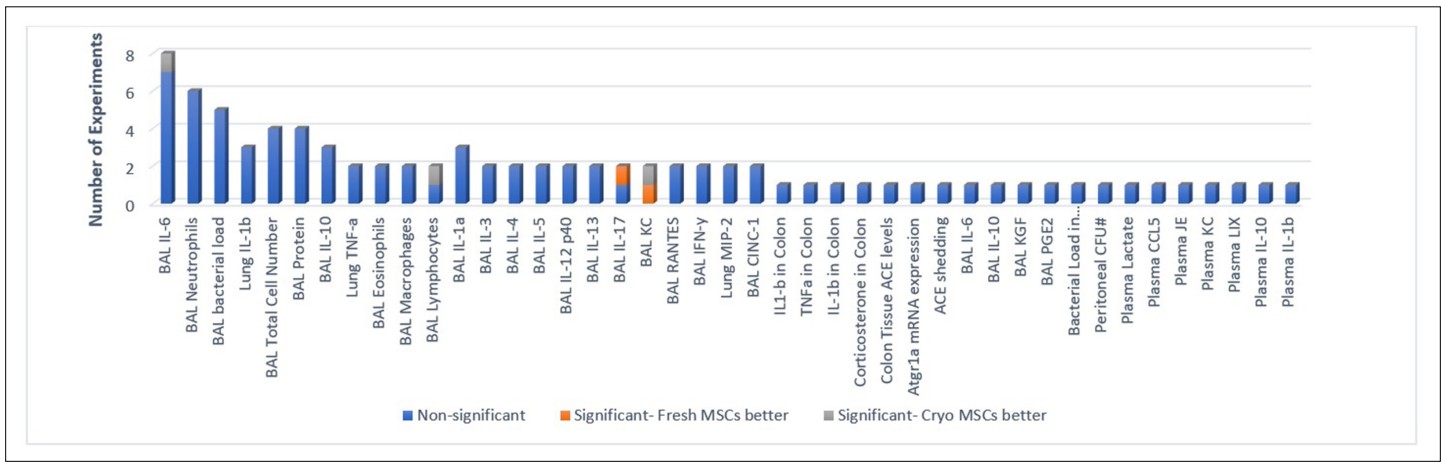

**Figure 3.** Primary in-vivo outcomes. All the outcomes related to protein (cytokine) expression and secretion are presented below. Number of experiments represent the number of separate comparisons between freshly cultured and cryopreserved MSCs on surrogate measures of in vivo efficacy.

supports reduced viability in cryopreserved versus freshly cultured MSCs, which is in keeping with previously published studies (*Eaker et al., 2013*; *Robb et al., 2019*).

Cryopreservation under safe and quality-controlled conditions remains critical for future real-world applications of MSC therapies (*Abazari et al., 2017*) by easing the logistical burden of supplying freshly cultured MSCs, enabling quality control and standardization of the cell preparation, and to facilitate the logistical transport of cellular products to hospitals. Some studies have shown that cryopreservation does not negatively impact MSCs; even if stored in cryopreservation for up to 23–24 years (*Shen et al., 2012*; *Badowski et al., 2014*; *Marquez-Curtis et al., 2015*). However, other studies have demonstrated mixed effects with both short-term and long-term cryopreservation (*Dariolli et al., 2013*; *Kotobuki et al., 2005*). Notably, most of these studies lack a clear assessment of MSC in vivo function. A recent systematic review of 41 in vitro studies that examined bone-marrow-derived MSCs (BM-MSCs) demonstrated that MSC cell morphology, marker expression, proliferation potential and tri-lineage differentiation capability were unaffected by stresses imposed by freezing and thawing, whereas viability, attachment to plasticware and migration, genomic stability and paracrine function of MSCs demonstrated conflicting results (*Bahsoun et al., 2019*). Out of their included 41 studies, only eight studied MSCs' immune function (88% conducted co-culture assays) post-thaw with four studies concluding a negative effect and four concluding no effect of cryopreservation on MSC in vitro immune function. Interestingly, this review found that the immediate post-thaw viability varied from about 50% to 100% among the included studies; 16 studies reported no change in viability immediately after thawing and 10 studies reported significantly lower viability (*Bahsoun et al., 2019*).

Cryopreserved MSCs have a higher percentage of apoptotic cells than MSCs from fresh cultures (*Haack-Sørensen and Kastrup, 2011*). Many factors could contribute to the diminished viability and functionality of cryopreserved MSCs, including the source of MSCs, rate of cooling, storage temperature and period, method of recovery from cryopreservation, and the cryoprotectants used (*Marquez-Curtis et al., 2015*). Cryopreserved MSCs are commonly frozen in 5–10% DMSO and or fetal bovine serum (FBS) (*Liu et al., 2010*; *Rowley et al., 1999*), but there are disadvantages of using these agents. DMSO is used extensively as a cryopreservation agent in the autologous hematopoietic stem cell transplant population and may be toxic at higher concentrations (*Alessandrino et al., 1999*). Adverse events have been associated with DMSO (most common are nausea, vomiting, weakness) (*Mitrus et al., 2018*) but a recent systematic review that examined safety of MSCs in randomized controlled trials (RCTs) found no serious adverse event safety signals for freshly cultured versus cryopreserved MSCs (*Thompson et al., 2020*). Furthermore, the use of animal proteins from FBS may theoretically increase the risk of transferring infectious agents or stimulating unwanted immunological responses. Despite the continued search for the most optimal cryoprotectant, no consensus has been developed on the safest type and concentration of cryoprotectant to use (*Galipeau and Sensébé, 2018*). Optimizing the rate of cooling is as important as the thawing process, both of which can further contribute to cell injury. Apoptotic and necrotic pathways are activated in these cells 6–48 h post-thaw in response to low temperature exposure (*Chinnadurai et al., 2016*; *Baust et al., 2009*). Remarkably, many studies demonstrate that MSCs, isolated from diverse sources, cryopreserved using various cooling rates, in the presence of different cryoprotectants, stored for various lengths of time, and at various sub-zero temperatures still retain their biological properties post-thaw except for viability (*Marquez-Curtis et al., 2015*). Viability of MSCs is considered an important indicator of cryopreservation success where at least 90% viability for fresh MSC product and 70% viability for cryopreserved MSC product are considered the benchmark for pre-clinical application (*Robb et al., 2019*). One provocative study found that recipient cytotoxic cell activity causing apoptosis of infused MSCs or infusion of ex-vivo apoptotic MSCs and suggested it is one of the proposed mechanisms of immunomodulation for MSCs and the lower viability (or increased number of apoptotic cells) may in fact play a positive role in reducing the host inflammatory state (*Galleu et al., 2017*). In a safety systematic review of MSC randomized trials, only 52% and 14.5% reported on viability and potency respectively (*Thompson et al., 2020*). Our systematic review also found that 13 of 18 included studies received an "unclear" risk of bias in 5 out of 10 domains of the SYRCLE risk of bias tool due to insufficient and unclear reporting of important variables (eg. cryopreservation process, storage conditions, blinding, etc.). Due to the importance of reporting risk of bias elements as well as the cryopreservation and thaw process that could impact MSC quantity, quality, and efficacy, interpretation of MSC research studies remains limited. We strongly encourage the standardized reporting of these parameters by

**Table 6.** In vitro outcomes where freshly cultured vs. cryopreserved MSCs were compared directly.

| Study | Outcome | Assay Used | Number (n) | Type and Source of MSCs | Time of cell preparation without MSC (hr) | Time of outcome measurement from MSC intervention (hr) | Concentration of MSCs | Pre-Treatment of MSCs | Negative Control (NC) | Positive Control (PC) | p-value for Fresh MSCs vs. control | p-value for Frozen MSCs vs. control | Fresh or Frozen MSC more effective? | p-value for Fresh vs. Frozen comparison |
|---|---|---|---|---|---|---|---|---|---|---|---|---|---|---|
| *Bárcia et al., 2017* | Viability | Trypan Blue | Fresh/Cultured (12); cryo <1 yr(12); cryo >3 yrs (5) | Human Umbilical Cord MSCs | N/A | 0 | NR | Fresh/Cultured MSCs were cryopreserved and then cultured for up to 5 days | N/A | N/A | N/A | N/A | ↔ | NS |
| | Apoptosis | Annexin V (and flow cytometry) | N/A | Human Umbilical Cord MSCs | N/A | 2 | NR | Fresh/Cultured MSCs were cryopreserved and then cultured for up to 5 days | N/A | Cultured cells incubated with $H_2O_2$ (2 mmol/L) for 2 hr | NR | NR | ↔ | NS |
| | Angiogenesis: Number of master junctions (branching points) | Matrigel/Human umbilical vein endothelial cell (HUVEC) tube formation assay | 2 | Human Umbilical Cord MSCs | 1 | 16 | $1 \times 10^6$ cells | Fresh/Cultured MSCs were cryopreserved and then cultured for up to 5 days; fresh and cryo co-cultured in basal media | N/A | HUVEC in Basal Media and HUVECs in Basal media with VEGF (100 ng/mL) | NR | NR | ↔ | NS |
| | Angiogenesis: segment/tube length | Matrigel/Human umbilical vein endothelial cell (HUVEC) tube formation assay | 2 | Human Umbilical Cord MSCs | 1 | 16 | $1 \times 10^6$ cells | Fresh/Cultured MSCs were cryopreserved and then cultured for up to 5 days; fresh and cryo co-cultured in basal media | N/A | HUVEC in Basal Media and HUVECs in Basal media with VEGF (100 ng/mL) | NR | NR | ↔ | NS |
| | Angiogenesis:total mesh area | Matrigel/Human umbilical vein endothelial cell (HUVEC) tube formation assay | 2 | Human Umbilical Cord MSCs | 1 | 16 | $1 \times 10^6$ cells | Fresh/Cultured MSCs were cryopreserved and then cultured for up to 5 days; fresh and cryo co-cultured in basal media | N/A | HUVEC in Basal Media and HUVECs in Basal media with VEGF (100 ng/mL) | NR | NR | ↔ | NS |
| *Gramlich et al., 2016* | Viability | TUNEL staining via Apo–Direct Apoptosis Detection Kit | 5 | Human MSCs | N/A | 24 | 30,000 MSCs | Both fresh and frozen cells were washed twice, resuspended in PBS and analyzed immediately or after 1 hr storage on wet ice | N/A | N/A | N/A | N/A | Fresh better | P<0.001 |
| | Viability | TUNEL staining via Apo–Direct Apoptosis Detection Kit | 5 | Human MSCs | N/A | 48 | 30,000 MSCs | Both fresh and frozen cells were washed twice, resuspended in PBS and analyzed immediately or after 1 hr storage on wet ice | N/A | N/A | N/A | N/A | Fresh better | P<0.001 |
| | Viability | TUNEL staining via Apo–Direct Apoptosis Detection Kit | 5 | Human MSCs | N/A | 72 | 30,000 MSCs | Both fresh and frozen cells were washed twice, resuspended in PBS and analyzed immediately or after 1 hr storage on wet ice | N/A | N/A | N/A | N/A | Fresh better | P=0.002 |
| | Metabolic Activity (measured by XXT) | XTT Assay | 6 | Human MSCs | N/A | 24 | 15,000 MSCs | N/A | N/A | N/A | N/A | N/A | ↔ | NS P=0.352 |

*Table 6 continued on next page*

*Table 6 continued*

| Measurement | Assay | n | Cell | | Time (h) | Cell number | | Control | | | Direction | Significance |
|---|---|---|---|---|---|---|---|---|---|---|---|---|
| Metabolic Activity (measured by XXT) | XTT Assay | 6 | Human MSCs | N/A | 48 | 15,000 MSCs | N/A | N/A | N/A | N/A | ↕ | NS P=0.312 |
| Metabolic Activity (measured by XXT) | XTT Assay | 6 | Human MSCs | N/A | 72 | 15,000MSCs | N/A | N/A | N/A | N/A | ↕ | NS P=0.971 |
| IDO activity: unstimulated MSC | Concentration of kynurenine in conditioned media | 6 | Human MSC | N/A | 48 | NR | N/A | N/A | N/A | N/A | ↕ | NS P=0.998 |
| IDO activity:MSC exposed to IFN-y | Concentration of kynurenine in conditioned media | 6 | Human MSC | N/A | 48 | NR | N/A | N/A | N/A | N/A | ↕ | NS P=0.099 |
| IDO activity: MSC exposed to IFN-y+TNF a | Concentration of kynurenine in conditioned media | 6 | Human MSC | N/A | 48 | NR | N/A | N/A | N/A | N/A | ↕ | NS P=0.951 |
| GDF-15: unstimulated | Human Growth Factor Array Q1 | 4 | Human MSC | N/A | 48 | 200,000 MSCs | N/A | Media Control | N/A | N/A | Frozen better | P=0.01 |
| GDF-15: stimulated with IFN-y/TNF-a | Human Growth Factor Array Q1 | 4 | Human MSC | N/A | 48 | 200,000 MSCs | N/A | Media Control | N/A | N/A | ↕ | NS P=0.99 |
| IGFBP-2: unstimulated | Human Growth Factor Array Q1 | 4 | Human MSC | N/A | 48 | 200,000 MSCs | N/A | Media Control | N/A | N/A | ↕ | NS P=0.32 |
| IGFBP-2: stimulated with IFN-y/TNF-a | Human Growth Factor Array Q1 | 4 | Human MSC | N/A | 48 | 200,000 MSCs | N/A | Media Control | N/A | N/A | ↕ | NS P=0.68 |
| IGFBP-3: unstimulated | Human Growth Factor Array Q1 | 4 | Human MSC | N/A | 48 | 200,000 MSCs | N/A | Media Control | N/A | N/A | ↕ | NS P=0.47 |
| IGFBP-3: stimulated with IFN-y/TNF-a | Human Growth Factor Array Q1 | 4 | Human MSC | N/A | 48 | 200,000 MSCs | N/A | Media Control | N/A | N/A | ↕ | NS P=0.75 |
| IGFBP-4: unstimulated | Human Growth Factor Array Q1 | 4 | Human MSC | N/A | 48 | 200,000 MSCs | N/A | Media Control | N/A | N/A | ↕ | NS P=0.39 |
| IGFBP-6: unstimulated | Human Growth Factor Array Q1 | 4 | Human MSC | N/A | 48 | 200,000 MSCs | N/A | Media Control | N/A | N/A | ↕ | NS P=0.69 |
| IGFBP-6: stimulated with IFN-y/TNF-a | Human Growth Factor Array Q1 | 4 | Human MSC | N/A | 48 | 200,000 MSCs | N/A | Media Control | N/A | N/A | Fresh better | P=0.03 |
| Insulin: stimulated with IFN-y/TNF-a | Human Growth Factor Array Q1 | 4 | Human MSC | N/A | 48 | 200,000 MSCs | N/A | Media Control | N/A | N/A | ↕ | NS P=0.71 |
| OPG: unstimulated | Human Growth Factor Array Q1 | 4 | Human MSC | N/A | 48 | 200,000 MSCs | N/A | Media Control | N/A | N/A | ↕ | NS P=0.39 |
| OPG: stimulated with IFN-y/TNF-a | Human Growth Factor Array Q1 | 4 | Human MSC | N/A | 48 | 200,000 MSCs | N/A | Media Control | N/A | N/A | ↕ | NS P=0.65 |
| PDGF-AA: unstimulated | Human Growth Factor Array Q1 | 4 | Human MSC | N/A | 48 | 200,000 MSCs | N/A | Media Control | N/A | N/A | ↕ | NS P=0.43 |
| PDGF-AA: stimulated with IFN-y/TNF-a | Human Growth Factor Array Q1 | 4 | Human MSC | N/A | 48 | 200,000 MSCs | N/A | Media Control | N/A | N/A | Frozen better | P=0.04 |
| PIGF: unstimulated | Human Growth Factor Array Q1 | 4 | Human MSC | N/A | 48 | 200,000 MSCs | N/A | Media Control | N/A | N/A | ↕ | NS P=0.83 |

*Table 6 continued on next page*

*Table 6 continued*

| | | | | | | | | | | | | | |
|---|---|---|---|---|---|---|---|---|---|---|---|---|---|
| SCF R: stimulated with IFN-γ/TNF-a | Human Growth Factor Array Q1 | 4 | Human MSC | N/A | 48 | 200,000 MSCs | N/A | N/A | Media Control | N/A | N/A | ↔ | NS P=0.06 |
| TGFb1: unstimulated | Human Growth Factor Array Q1 | 4 | Human MSC | N/A | 48 | 200,000 MSCs | N/A | N/A | Media Control | N/A | N/A | N/A | N/A |
| TGFb1: stimulated with IFN-γ/TNF-a | Human Growth Factor Array Q1 | 4 | Human MSC | N/A | 48 | 200,000 MSCs | N/A | N/A | Media Control | N/A | N/A | Fresh better | P=0.05 |
| VEGF: unstimulated | Human Growth Factor Array Q1 | 4 | Human MSC | N/A | 48 | 200,000 MSCs | N/A | N/A | Media Control | N/A | N/A | ↔ | NS P=0.30 |
| VEGF: stimulated with IFN-γ/TNF-a | Human Growth Factor Array Q1 | 4 | Human MSC | N/A | 48 | 200,000 MSCs | N/A | N/A | Media Control | N/A | N/A | ↔ | NS P=0.96 |
| **Tan et al., 2019** | | | | | | | | | | | | | |
| Viability | Trypan Blue | NR | Human BM | N/A | 0 | NR | N/A | N/A | N/A | N/A | N/A | ↔ | NS |
| Viability | Trypan Blue | NR | Human BM | N/A | 2 | NR | N/A | N/A | N/A | N/A | N/A | Fresh better | P<0.05 |
| Viability | Trypan Blue | NR | Human BM | N/A | 4 | NR | N/A | N/A | N/A | N/A | N/A | ↔ | NS |
| Viability | Trypan Blue | NR | Human BM | N/A | 6 | NR | N/A | N/A | N/A | N/A | N/A | ↔ | NS |
| Viability (Viable Cells) | Annexin V+Propidium iodide (AV/PI) | NR | Human BM | N/A | 0 | NR | N/A | N/A | N/A | N/A | N/A | ↔ | NS |
| Viability (Viable Cells) | Annexin V+Propidium iodide (AV/PI) | NR | Human BM | N/A | 2 | NR | N/A | N/A | N/A | N/A | N/A | ↔ | NS |
| Viability (Viable Cells) | Annexin V+Propidium iodide (AV/PI) | NR | Human BM | N/A | 4 | NR | N/A | N/A | N/A | N/A | N/A | ↔ | NS |
| Viability (Viable Cells) | Annexin V+Propidium iodide (AV/PI) | NR | Human BM | N/A | 6 | NR | N/A | N/A | N/A | N/A | N/A | Fresh better | P<0.05 |
| Viability(Early apoptotic cells) | Annexin V+Propidium iodide (AV/PI) | NR | Human BM | N/A | 0 | NR | N/A | N/A | N/A | N/A | N/A | ↔ | NS |
| Viability(Early apoptotic cells) | Annexin V+Propidium iodide (AV/PI) | NR | Human BM | N/A | 2 | NR | N/A | N/A | N/A | N/A | N/A | ↔ | NS |
| Viability(Early apoptotic cells) | Annexin V+Propidium iodide (AV/PI) | NR | Human BM | N/A | 4 | NR | N/A | N/A | N/A | N/A | N/A | ↔ | NS |
| Viability(Early apoptotic cells) | Annexin V+Propidium iodide (AV/PI) | NR | Human BM | N/A | 6 | NR | N/A | N/A | N/A | N/A | N/A | Fresh better | P<0.05 |
| Viability (Late apoptotic cells) | Annexin V+Propidium iodide (AV/PI) | NR | Human BM | N/A | 0 | NR | N/A | N/A | N/A | N/A | N/A | ↔ | NS |
| Viability (Late apoptotic cells) | Annexin V+Propidium iodide (AV/PI) | NR | Human BM | N/A | 2 | NR | N/A | N/A | N/A | N/A | N/A | ↔ | NS |
| Viability (Late apoptotic cells) | Annexin V+Propidium iodide (AV/PI) | NR | Human BM | N/A | 4 | NR | N/A | N/A | N/A | N/A | N/A | Fresh better | P<0.05 |

*Table 6 continued on next page*

*Table 6 continued*

| | Method | | Source | | Time (hr) | Density | Processing | Control | Treatment | | | | Trend | Comparison |
|---|---|---|---|---|---|---|---|---|---|---|---|---|---|---|
| Viability (Late apoptotic cells) | Annexin V+Propidium iodide (AV/PI) | NR | Human BM | N/A | 6 | NR | N/A | N/A | N/A | N/A | N/A | N/A | | Fresh better $P<0.05$ |
| Phagocytosis | PBMCs pre-treated with LPS the co-culture with MSC at ratio of 1:5 for 24 hr | 3–6 | Human BM MSC: Donor 1 | N/A | 24 | NR | N/A | Naive PBMC | LPS treated PBMC | PC: $P<0.0001$ | N/A | PC: $P<0.0001$ | ↕ | NS |
| Phagocytosis | PBMCs pre-treated with LPS the co-culture with MSC at ratio of 1:5 for 24 hr | 3–6 | Human BM MSC: Donor 2 | N/A | 24 | NR | N/A | Naive PBMC | LPS treated PBMC | NS | NS | | ↕ | NS |
| Phagocytosis | PBMCs pre-treated with LPS the co-culture with MSC at ratio of 1:5 for 24 hr | 3–6 | Human BM MSC: Donor 3 | N/A | 24 | NR | N/A | Naive PBMC | LPS treated PBMC | PC: $P<0.001$ | PC: $P<0.001$ | | ↕ | NS |
| Permeability | Endothelial cell (EC) treated with LPS for 6 hr then co-culture with MSC for 24 hr at ratio of 1:2 followed by adding FITC-dextran to the transwell insert | NR | Human BM MSC: Donor 1 | N/A | 24 | NR | N/A | Non-treated EC | LPS treated EC | PC: $P<0.01$ | PC: $P<0.01$ | | ↕ | NS |
| Permeability | Endothelial cell (EC) treated with LPS for 6 hr then co-culture with MSC for 24 hr at ratio of 1:2 followed by adding FITC-dextran to the transwell insert | NR | Human BM MSC: Donor 2 | N/A | 24 | NR | N/A | Non-treated EC | LPS treated EC | PC: $P<0.01$ | PC: $P<0.01$ | | ↕ | NS |
| Permeability | Endothelial cell (EC) treated with LPS for 6 hr then co-culture with MSC for 24 hr at ratio of 1:2 followed by adding FITC-dextran to the transwell insert | NR | Human BM MSC: Donor 3 | N/A | 24 | NR | N/A | Non-treated EC | LPS treated EC | PC: $P<0.001$ | PC: $P<0.001$ | | ↕ | NS |
| **Bharti et al., 2020** Growth Curve | Countess automated cell counter | NR | Canine BM | N/A | 24 | $1 \times 10^4$ cells/ml | Frozen cells were thawed in distilled water at 36 °C for 45–60 s then enzymatically detached from mesh and added in re-warmed media with 15% FBS and washed twice at 1200 rpm for 5 min | N/A | N/A | N/A | N/A | N/A | | NS |

*Table 6 continued on next page*

*Table 6 continued*

| | | | | | | | | | | | | | |
|---|---|---|---|---|---|---|---|---|---|---|---|---|---|
| Growth Curve | Countess automated cell counter | NR | Canine BM | N/A | 48 | $1 \times 10^4$ cells/ml | Frozen cells were thawed in distilled water at 36 °C for 45–60 s then enzymatically detached from mesh and added in re-warmed media with 15% FBS and washed twice at 1200 rpm for 5 min | N/A | N/A | N/A | N/A | ↕ | NS |
| Growth Curve | Countess automated cell counter | NR | Canine BM | N/A | 72 | $1 \times 10^4$ cells/ml | Frozen cells were thawed in distilled water at 36 °C for 45–60 s then enzymatically detached from mesh and added in re-warmed media with 15% FBS and washed twice at 1200 rpm for 5 min | N/A | N/A | N/A | N/A | ↕ | NS |
| Growth Curve | Countess automated cell counter | NR | Canine BM | N/A | 96 | $1 \times 10^4$ cells/ml | Frozen cells were thawed in distilled water at 36 °C for 45–60 s then enzymatically detached from mesh and added in re-warmed media with 15% FBS and washed twice at 1200 rpm for 5 min | N/A | N/A | N/A | N/A | ↕ | NS |
| Growth Curve | Countess automated cell counter | NR | Canine BM | N/A | 120 | $1 \times 10^4$ cells/ml | Frozen cells were thawed in distilled water at 36 °C for 45–60 s then enzymatically detached from mesh and added in re-warmed media with 15% FBS and washed twice at 1200 rpm for 5 min | N/A | N/A | N/A | N/A | ↕ | NS |
| Growth Curve | Countess automated cell counter | NR | Canine BM | N/A | 144 | $1 \times 10^4$ cells/ml | Frozen cells were thawed in distilled water at 36 °C for 45–60 s then enzymatically detached from mesh and added in re-warmed media with 15% FBS and washed twice at 1200 rpm for 5 min | N/A | N/A | N/A | N/A | ↕ | NS |
| Growth Curve | Countess automated cell counter | NR | Canine BM | N/A | 168 | $1 \times 10^4$ cells/ml | Frozen cells were thawed in distilled water at 36 °C for 45–60 s then enzymatically detached from mesh and added in re-warmed media with 15% FBS and washed twice at 1200 rpm for 5 min | N/A | N/A | N/A | N/A | ↕ | NS |

*Table 6 continued on next page*

*Table 6 continued*

| | | | | | | | | | | | | |
|---|---|---|---|---|---|---|---|---|---|---|---|---|
| Growth Curve | Countess automated cell counter | NR | Canine BM | N/A | 192 | 1 × 10⁴ cells/ml | Frozen cells were thawed in distilled water at 36 °C for 45–60 s then enzymatically detached from mesh and added in re-warmed media with 15% FBS and washed twice at 1200 rpm for 5 min | N/A | N/A | N/A | ↕ | NS |
| Growth Curve | Countess automated cell counter | NR | Canine BM | N/A | 216 | 1 × 10⁴ cells/ml | Frozen cells were thawed in distilled water at 36 °C for 45–60 s then enzymatically detached from mesh and added in re-warmed media with 15% FBS and washed twice at 1200 rpm for 5 min | N/A | N/A | N/A | ↕ | NS |
| Growth Curve | Countess automated cell counter | NR | Canine BM | N/A | 240 | 1 × 10⁴ cells/ml | Frozen cells were thawed in distilled water at 36 °C for 45–60 s then enzymatically detached from mesh and added in re-warmed media with 15% FBS and washed twice at 1200 rpm for 5 min | N/A | N/A | N/A | ↕ | NS |
| Growth Curve | Countess automated cell counter | NR | Canine BM | N/A | 264 | 1 × 10⁴ cells/ml | Frozen cells were thawed in distilled water at 36 °C for 45–60 s then enzymatically detached from mesh and added in re-warmed media with 15% FBS and washed twice at 1200 rpm for 5 min | N/A | N/A | N/A | ↕ | NS |
| Growth Curve | Countess automated cell counter | NR | Canine BM | N/A | 288 | 1 × 10⁴ cells/ml | Frozen cells were thawed in distilled water at 36 °C for 45–60 s then enzymatically detached from mesh and added in re-warmed media with 15% FBS and washed twice at 1200 rpm for 5 min | N/A | N/A | N/A | ↕ | NS |
| Growth Curve | Countess automated cell counter | NR | Canine BM | N/A | 312 | 1 × 10⁴ cells/ml | Frozen cells were thawed in distilled water at 36 °C for 45–60 s then enzymatically detached from mesh and added in re-warmed media with 15% FBS and washed twice at 1200 rpm for 5 min | N/A | N/A | N/A | ↕ | NS |

*Table 6 continued on next page*

*Table 6 continued*

| Study | Outcome measure | Assay / method | n | Cell source | | Time | Cell concentration | Intervention / culture condition | | | | Result | Significance |
|---|---|---|---|---|---|---|---|---|---|---|---|---|---|
| | CD 105 expression | Antibody assay | NR | Canine BM | N/A | Overnight | NR | Primary antibodies (1:100 dilutions) were used for localizing different markers (CD73, CD90, CD105, CD34) with an overnight incubation period at 4 °C. | N/A | N/A | N/A | ↕ | NS |
| | CD 90 expression | Antibody assay | NR | Canine BM | N/A | Overnight | NR | Primary antibodies (1:100 dilutions) were used for localizing different markers (CD73, CD90, CD105, CD34) with an overnight incubation period at 4 °C. | N/A | N/A | N/A | ↕ | NS |
| | CD 73 expression | Antibody assay | NR | Canine BM | N/A | Overnight | NR | Primary antibodies (1:100 dilutions) were used for localizing different markers (CD73, CD90, CD105, CD34) with an overnight incubation period at 4 °C. | N/A | N/A | N/A | ↕ | NS |
| | Population Doubling Time | N/A | NR | Canine BM | N/A | N/A | $1 \times 10^4$ cells/ml | N/A | N/A | N/A | N/A | ↕ | NS |
| **Rogulska et al., 2019** | Metabolic Activity/Proliferation rate | Alamar Blue | 3 | Human Adipose | N/A | 48 | NR | MSCs culture in PS1D-based gel | N/A | N/A | N/A | Fresh better | P<0.05 |
| | Metabolic Activity/Proliferation rate | Alamar Blue | 3 | Human Adipose | N/A | 96 | NR | MSCs culture in PS1D-based gel | N/A | N/A | N/A | Fresh better | P<0.05 |
| | Metabolic Activity/Proliferation rate | Alamar Blue | 3 | Human Adipose | N/A | 144 | NR | MSCs culture in PS1D-based gel | N/A | N/A | N/A | ↕ | NS |
| | Viability | Alamar Blue | 3 | Human Adipose | N/A | 24 | NR | N/A | N/A | N/A | N/A | Fresh better | P<0.05 |
| **Khan et al., 2019** | Antioxidant Concentration (2 fresh groups:GFP-MSC and HO-1 MSC) | Antioxidant Assay | 6 | Canine adipose | NR | NR | NR | Lentivirus-mediated GFP and HO-1 gene insertion into Ad-MSCs | N/A | N/A | N/A | Fresh better | P<0.05 |
| | Viability | Trypan Blue | 6 | Human Umbilical Cord | 0 | 0, 2, 4, 24, 48 hr | $1 \times 10^4$ cells/well | None | N/A | N/A | N/A | ↕ | NS |
| | Viability | Water-soluble tetrazolium salt (WST) assay | 6 | Human Umbilical Cord | 0 | 0, 2, 4, 24, 48 hr | $1 \times 10^4$ cells/well | None | N/A | N/A | N/A | ↕ | NS |
| **Yea et al., 2020** | Population Doubling Time | Cell counting | 6 | Human Umbilical Cord | 0 | 4, 8, 12, 16, 20 days | $3 \times 10^3$ cells/cm$^2$ | None | N/A | N/A | N/A | ↕ | NS |
| **Horiuchi et al., 2021** | Bioluminescence | IVIS Lumina XRMS series III instrument (SPI, Tokyo, Japan) | 4 | Rat Synovial MSCs | 0 | Same day | Varying concentrations | None | N/A | N/A | N/A | ↕ | NS |

N/A = Not applicable (e.g. if the experiment set up did not include a particular variable). NR = Not reported (e.g. if a particular variable was part of the experiment set up but not explicitly reported on in results section or graph).

authors, reviewers, and journal editors as markers of reporting quality and to enhance transparency, reproducibility, and interpretation of MSC research studies.

From the perspective of clinical research and potential efficacy of cryopreserved MSCs, a phase III randomized clinical trial that examined whether a cryopreserved MSC product, PROCHYMAL (Reme-stemcel-L), or placebo compared to standard second line therapies alone in children with acute graft-versus-host disease (aGVHD) showed that high risk patients were more likely to have a partial response at 28 days with Remestemcel. Furthermore, a recently published systematic review that examined 55 randomized trials which used a MSC product versus control/usual care not only suggested evidence for safety of cryopreserved MSCs but also potential efficacy. Of the 15 trials that studied a cryopre-served product, 5 of them (33%) found significant differences favoring cryopreserved MSCs in either the primary or secondary endpoints (*Kebriaei et al., 2020*).

There are several strengths in this current systematic review. First, we have published our protocol which includes a transparent search strategy, pre-defined classifications for cryopreserved and freshly cultured MSCs and outcome measures, and minimal exclusion criteria. Ours is the first comprehensive systematic review assessing the in vivo efficacy of cryopreserved MSCs when directly compared to freshly cultured MSCs in animal models of inflammation. All variables and experimental details were collected and summarized systematically. Given the breadth and variety of in vivo and in vitro outcome measures, we report our data by considering each experiment where cryopreserved and freshly cultured MSCs are compared as an individual hypothesis test. Our review provides the totality of the existing pre-clinical evidence base, and we hope it will provide additional rationale for considering a cryopreserved MSC product for use in pre-clinical studies and clinical trials, and help identify research gaps for future related research (*Galipeau and Sensébé, 2018*).

Our study did have some limitations. Given our emphasis on including studies that examined MSC in vivo efficacy, we excluded all studies that only conducted in vitro studies. This led to a significant number of cryopreserved MSC studies being excluded and hence, our in vitro outcome reporting may be incomplete. However, when considering whether cryopreserved MSCs may be efficacious in clinical settings, pre-clinical in vivo efficacy outcomes might be more convincing than in vitro studies alone. Most of the preclinical studies did not provide sufficient information to adequately perform the SYRCLE risk of bias assessment, resulting in unclear reporting in at least three bias domains or more in all but one study, despite our attempts to contact authors to obtain further study details. Our ability to conduct meta-analyses on our primary outcome measures and according to subgroups was significantly limited by the heterogeneity of animal models included and breadth of outcomes measured. Finally, it is possible that other important in vivo pre-clinical efficacy or in vitro potency outcomes were not reported in our review. However, we designed and then conducted a systematic and transparent search using a pre-published protocol to enhance transparency and reproducibility, and to ensure we captured the totality of the evidence according to our study question. Questions remain related to MSC mechanisms of action in response to different immune stimuli, such as the effect of xenotransplanation. Further research to understand where there may be differences in effects of syngeneic MSCs as compared to xenogenic MSCs in models of inflammatory diseases related to HLA stimulation/expression, co-stimulatory molecules, paracrine factors, and species-specific cytokines and receptors may assist successful translation in human clinical trials (*Prockop and Lee, 2017*). Our review reported pre-dominantly on different biological outcome measures which does not provide a measure of overall animal health in a given inflammatory animal model. However, certain biological outcomes may be part of the mechanistic/causal pathway related to the disease (in the animal and humans) and may be considered as important surrogates for overall health. These biological outcomes in pre-clinical studies may also help to inform the exploration of them as predictive or prognostic variables in human clinical trials.

## Conclusions

Our study provides a comprehensive systematic review of pre-clinical studies comparing cryopreserved versus freshly cultured MSCs in animal models of inflammation. Our findings suggest that for the majority of outcomes measured in this review, cryopreservation does not negatively impact in vivo efficacy or in vitro potency of MSCs. With our systematic summary of the current evidence base, we hope it may provide MSC basic and research scientists additional rationale for considering a cryopreserved MSC product for use in pre-clinical studies and clinical trials, and help identify research gaps

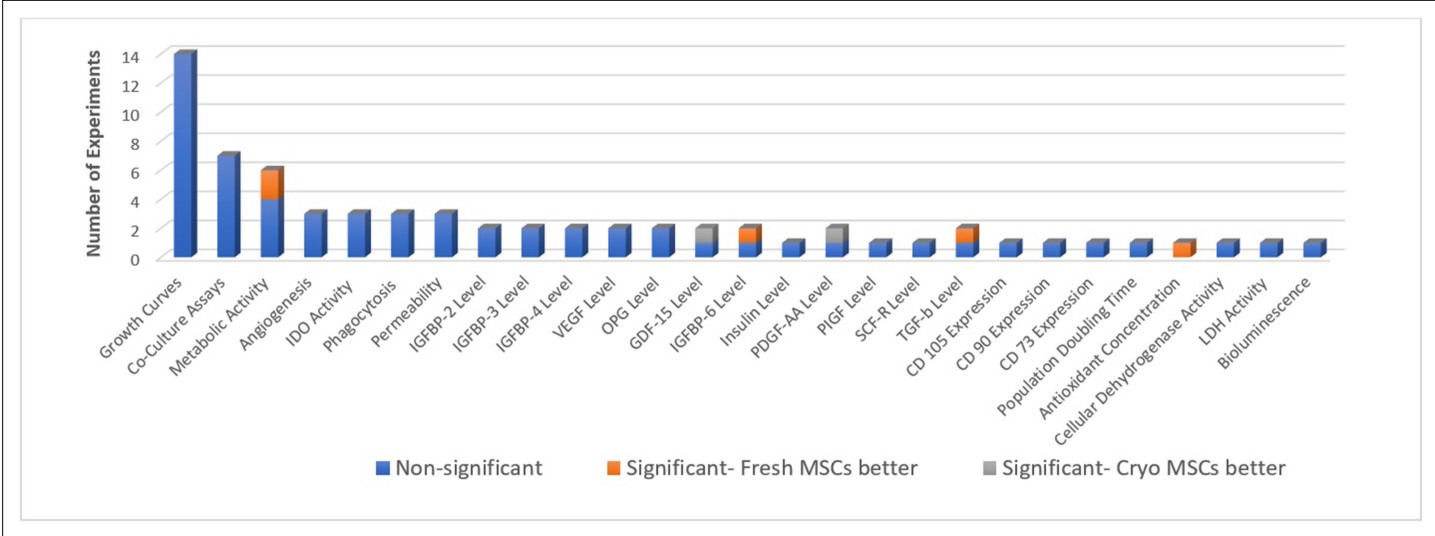

**Figure 4.** In-vitro potency outcomes. All the in-vitro reported outcomes are displayed below. Number of experiments represent the number of separate comparisons between freshly-cultured and cryopreserved MSCs on surrogate measures of in vivo efficacy.

for future MSC-related research. We also strongly encourage the standardized reporting of important parameters related to risk of bias, MSC processing characteristics (e.g. cryopreservation and thawing protocols), storage conditions, viability, and potency as markers of study quality and to enhance transparency, reproducibility, and interpretation of MSC research studies.

## Acknowledgements

We acknowledge Emily Doxtator for her help in the initial screening of studies, Risa Shorr for her help in conducting the scientific, comprehensive search strategy, and Diana Wolfe for her contribution to the development of the protocol for this systematic review. We acknowledge the Ontario Institute for Regenerative Medicine and the Stem Cell Network for funding this systematic review. We also acknowledge the following authors who responded to our request for further information: Taru Sharma, Shahd Horie, Daniel O'Toole, and James Ankrum.

## Additional information

### Funding

| Funder | Grant reference number | Author |
|---|---|---|
| Ontario Institute for Regenerative Medicine | 2016-0147 | Chintan Dave |
| Stem Cell Network | | Lauralyn McIntyre |

The funders had no role in study design, data collection and interpretation, or the decision to submit the work for publication.

### Author contributions

Chintan Dave, Conceptualization, Data curation, Formal analysis, Investigation, Methodology, Project administration, Supervision, Visualization, Writing – original draft, Writing – review and editing; Shirley HJ Mei, Conceptualization, Data curation, Formal analysis, Writing – review and editing; Andrea McRae, Katrina J Sullivan, Data curation, Formal analysis, Writing – review and editing; Christine Hum, Data curation, Formal analysis, Investigation, Visualization, Writing – original draft, Writing – review and editing; Josee Champagne, Conceptualization, Data curation, Methodology, Visualization,

**Table 7.** Summary of all in vitro PBMC Proliferation assays from included studies.

| Study | MSCs Used | Solution | Addition to solution | Responder Cells | Fresh vs. Frozen Comparison | Duration of Culture | Proliferation Measurement | Ratio (MSC:Responder Cells) 1:1 | 1:3 | 1:6 | 1:10 | 1:12 | 1:50 |
|---|---|---|---|---|---|---|---|---|---|---|---|---|---|
| *Bárcia et al., 2017* | Cultured and Freshly-thawed MSCs were irradiated with 50 Gy prior to use | RPMI | 5% HEPES, 5% Pen-Strep, 5% NaPyr and 5% human serum | PBMC stimulated with anti-CD3, anti-CD28, and IL-2. | Yes | 16 hr | Percentage of T cells proliferation/ suppression | Yes | | | | | Yes |
| *Gramlich et al., 2016* | Cultured and Freshly-thawed MSCs | RPMI | 10% (v/v) FBS, 1% (v/v) Penicillin/ Streptomycin, and 1% (v/v) L-glutamine | PBMC stimulated with 250,000 Human T-activator CD3+/D28+Dynabeads | Yes | 144 hr | CFSE Cell Proliferation Kit | | Yes | Yes | Yes | Yes | |
| *Tan et al., 2019* | Cultured and Freshly-thawed MSCs | NR | NR | PBMC stimulated with Dynabeads Human T-Activator CD3/CD28 | Yes | 120 hr | | | Yes | | | | |

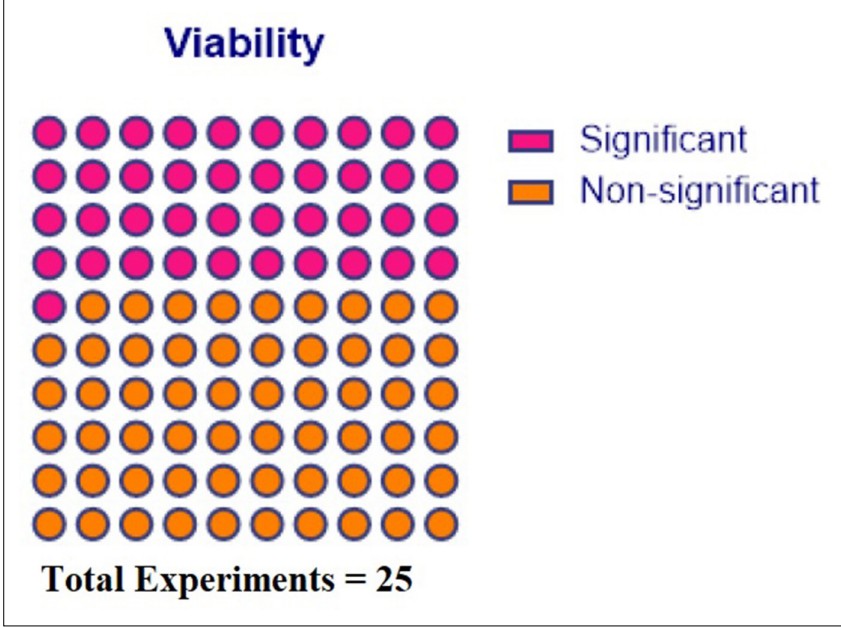

**Figure 5.** Comparison of viability. Experiments where viability at varying time points of freshly-cultured and cryopreserved MSCs were compared directly are presented below.

Writing – review and editing; Tim Ramsay, Conceptualization, Methodology, Visualization, Writing – review and editing; Lauralyn McIntyre, Conceptualization, Data curation, Formal analysis, Funding acquisition, Methodology, Project administration, Supervision, Writing – original draft, Writing – review and editing

#### Author ORCIDs
Chintan Dave http://orcid.org/0000-0003-4371-7645
Tim Ramsay http://orcid.org/0000-0001-8478-8170
Lauralyn McIntyre http://orcid.org/0000-0001-7421-1407

#### Decision letter and Author response
Decision letter https://doi.org/10.7554/eLife.75053.sa1
Author response https://doi.org/10.7554/eLife.75053.sa2

## Additional files

#### Supplementary files
• Supplementary file 1. Search strategy.
• Supplementary file 2. PRISMA checklist.
• Supplementary file 3. AGREE-II tool prompting questions.
• Supplementary file 4. Data collection items.
• MDAR checklist

#### Data availability
All data generated or analyzed in our review are provided in the attached tables and figures.

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
