## [Editor Report]

The pre-clinical systematic review by Dave C et al. covers an important and highly debated topic, which is the advantages and disadvantages of the use of freshly cultured vs cryopreserved mesenchymal stromal cells (MSCs). The authors conduct an appropriate survey and bias analysis and focus their review on reported studies on animal models of inflammation. They conclude that there are no significant differences between freshly-isolated or cryopreserved MSCs in terms of their pre-clinical efficacy.

---

## [Decision Letter]

**Decision letter after peer review:**

Thank you for submitting your article "Comparison of Freshly-Cultured versus Cryopreserved Mesenchymal Stem Cells in Animal Models of Inflammation: A Pre-Clinical Systematic Review" for consideration by *eLife*. Your article has been reviewed by 3 peer reviewers, one of whom is a member of our Board of Reviewing Editors, and the evaluation has been overseen by a Reviewing Editor and Mone Zaidi as the Senior Editor. The reviewers have opted to remain anonymous.

Essential revisions:

1) Extend their systematic review to make it more comprehensive and obtain higher power of analysis. If the heterogeneity of studies and parameters available does not allow to perform cross-comparisons, the limitations of the conclusions and interpretations should be acknowledged.

2) Key parameters that were not available for all studies are the methods for cryopreservation and thawing before use. These have a direct impact on cell viability and activity and, if not available, the conclusions should be toned down.

3) Previous studies directly comparing freshly-culture vs cryopreserved MSCs or demonstrating impaired immunosuppressive properties (e.g. 10.3109/14653249.2011.623691 or 10.1002/stem.2415) are not cited or discussed.

4) The authors discuss the differential level of apoptosis (higher in cryopreserved MSCs, compared with freshly cultured cells). The possible role of apoptosis in the immunodulatory effects of MSCs (doi: 10.1126/scitranslmed.aam7828) should be discussed as well.

5) The authors should include studies administering human MSCs using immuno-compromised animals to avoid immune rejection. The rationale for excluding this group is not justified.

6) Further clarification on criteria used to assess the methodological rigor of the included studies is needed.

7) The impact of cryopreservation compared with many other experimental variables, such as route of administration, cell dose, cell sources, etc. should be discussed. The various administration routes should be considered particularly in the context of treatment of inflammatory diseases in pre-clinical animal models. The authors should acknowledge that reporting on differences for a single variable is unlikely to provide a measure of overall animal health and provide clinically meaningful information of the duration and severity of the disease course. The authors should place more emphasis on this point in the discussion and acknowledge pitfalls associated with focusing on any one outcome metric.

8) The impact of the study would be greatly improved if a meta-analysis was performed including a higher number of studies. More recent studies need to be included as well.

*Reviewer #2 (Recommendations for the authors):*

The manuscript by Dave et al., describes a systemic analysis of existing literature comparing the potency and efficacy of fresh vs. cryopreserved mesenchymal stem/stromal cells (MSCs) in preclinical disease models. The systemic analysis is properly executed and appropriately evaluates biases in reporting. Inclusion and exclusion criteria are well described and appear to be well justified. However, the studies included in the analysis are broad based with respect to disease type, cell source, route of administration, dose, and outcome measures used to evaluate efficacy are also highly variable. Due to this high degree of heterogeneity, a rigorous meta-analysis was not feasible. Therefore, the results of the study are limited in scope, and it is unclear if the conclusions of the study are clinically meaningful. Therefore, aspects of the study require clarification.

1. The literature search was not biased by language and the authors were careful to encompass the broad terminology used to describe MSCs in their search, which ensures all relevant studies were captured. Despite these efforts, only 15 studies were included and these encompassed an array of conditions, such as lung injury, sepsis, allergic airway inflammation, wound healing, neurological and ocular disease. Therefore, while the study addresses a focused question of clinical relevance, the analysis of such heterogeneous studies limits overall confidence in the reliability of the results.

2. The methods section does a good job of defining inclusion and exclusion criteria. However, if xenogeneic, syngeneic, autologous and allogeneic MSCs were included it is unclear why use of immuno-compromised animals was excluded. For example, studies administering human MSCs to mice often employ immuno-compromised animals to avoid immune rejection. The authors should explain their rationale for excluding this group.

3. Bias assessment included criteria related to reporting bias. However, since it is critical to exclude studies that employ inferior methodological approaches, the authors should provide further clarification on criteria used to assess the methodological rigor of the included studies.

4. The authors indicate that the duration of cryopreservation was not reported in half of the studies analyzed, and the remaining studies used varying durations. Since the authors do not rule out cryopreservation duration as a variable that influences potency/efficacy, the inconstancy between studies is a concern. This, coupled with the fact that variable methods were used to thaw cells, and viability of the final product varied significantly (60-97%), raises serious concerns about the reliability of the analysis. These concerns regarding cryopreservation are also relevant for many other variables, such as route of administration, cell dose, cell sources, etc.

5. The impact of the study would be improved if a meta-analysis was performed. However, due to the heterogeneity of the studies evaluated, differences between fresh vs. cryopreserved MSCs were converted to mean and standard deviation for each outcome measure, which is of limited significance. For example, reporting on differences for a single variable is unlikely to provide a measure of overall animal health and provide clinically meaningful information of the duration and severity of the disease course. The authors should place more emphasis on this point in the discussion and acknowledge pitfalls associated with focusing on any one outcome metric.

*Reviewer #3 (Recommendations for the authors):*

In the present review, Dave et al., provide an innovative and eye-catching comparison on the efficacy between freshly-cultured and cryopreserved mesenchymal stem cells in the pre-clinical treatment of inflammatory diseases. As the authors focused on detailed analysis of both in-vivo pre-clinical and in-vitro MSC potency outcomes from those published studies until June, 2020, no significant difference was detected in pre-clinical efficacy between freshly-cultured and cryopreserved MSCs basing on a massive electronic search and bioinformatics analysis on the on-line literature databases. This review is well-structured and provide evidence for considering a cryopreserved MSC product in further clinical trials.

The various administration routes should be detailed analyzed for treating inflammatory diseases in pre-clinical animal models. Besides, more recent published studies in 2021 should be included.

---

## [Author Response]

Essential revisions:1) Extend their systematic review to make it more comprehensive and obtain higher power of analysis. If the heterogeneity of studies and parameters available does not allow to perform cross-comparisons, the limitations of the conclusions and interpretations should be acknowledged.

Thank you. We have updated our search strategy to include all published studies that met our pre-specified eligibility criteria in our published protocol until January 13, 2022. We screened 775 additional studies; 3 of them met our inclusion/exclusion criteria and were included in our final analysis. We have included a line in the Results section which explains why we were not able to perform meta-analyses and have also added this limitation in the limitations paragraph in the Discussion section (Page 13 and Page 23).

2) Key parameters that were not available for all studies are the methods for cryopreservation and thawing before use. These have a direct impact on cell viability and activity and, if not available, the conclusions should be toned down.

Thank you for this comment. We agree that these are important parameters that may affect cell viability and function as we discussed in the Discussion section of the manuscript. We contacted authors to obtain additional information that was not reported in their studies. We have now also added this important point as a limitation in the limitations section (Page 22, Lines 340-342) and in the conclusion. We also further advise that reviewers and journal editors mandate these parameters as standardized reporting items in publications to enhance transparency, reproducibility, and the conclusions that can be drawn from these studies (in vitro, pre-clinical animal, and human clinical trials) (Page 21).

3) Previous studies directly comparing freshly-culture vs cryopreserved MSCs or demonstrating impaired immunosuppressive properties (e.g. 10.3109/14653249.2011.623691 or 10.1002/stem.2415) are not cited or discussed.

Thank you for this comment. These studies are notable within the field of MSC research and demonstrate important *in-vitro* findings that relate to impairment of MSC immunosuppressive properties due to cryopreservation. Our systematic review reports on *in-vitro* potency measures in studies of *in-vivo* models of inflammation as per the eligibility criteria in our published protocol (1). We have now cited these *in-vitr*o studies in the introduction section of our manuscript (Page 4).

4) The authors discuss the differential level of apoptosis (higher in cryopreserved MSCs, compared with freshly cultured cells). The possible role of apoptosis in the immunodulatory effects of MSCs (doi: 10.1126/scitranslmed.aam7828) should be discussed as well.

Thank- you. We have added the findings from this provocative study to our Discussion section (Page 20).

5) The authors should include studies administering human MSCs using immuno-compromised animals to avoid immune rejection. The rationale for excluding this group is not justified.

We excluded these animal models a priori from our pre-defined eligibility criteria in our published protocol because our primary aim was to examine the efficacy of cryopreserved versus freshly-cultured MSCs on measures of inflammation in animal models with an intact immune system. Furthermore, an intact immune system may be required for MSC immunomodulation via the host cytotoxic cell activity (2). We have added this information as the rationale for excluding these animals from the systematic review in the exclusion section (Page 7) of the manuscript.

6) Further clarification on criteria used to assess the methodological rigor of the included studies is needed.

SYRCLE risk of bias tool for animal studies contains 10 entries and are related to selection bias, performance bias, detection bias, attrition bias, reporting bias and other biases. Half these items are in agreement with the items in the Cochrane risk of bias tool. Most of the variations between the two tools are due to differences in design between RCTs and animal studies. To provide further clarification on the criteria used to assess the methodological rigor, we have included an additional table in the Supplementary files (Supplementary Table 2) that outlines the signaling questions employed to reach decisions about risk of bias within the 10 domains.

7) The impact of cryopreservation compared with many other experimental variables, such as route of administration, cell dose, cell sources, etc. should be discussed. The various administration routes should be considered particularly in the context of treatment of inflammatory diseases in pre-clinical animal models. The authors should acknowledge that reporting on differences for a single variable is unlikely to provide a measure of overall animal health and provide clinically meaningful information of the duration and severity of the disease course. The authors should place more emphasis on this point in the discussion and acknowledge pitfalls associated with focusing on any one outcome metric.

We were unable to perform several of the sub-groups that we had planned *apriori* in our published protocol due to the heterogeneity of the included studies and the lack of comprehensive reporting. We have now added further mention to this point in our results and Discussion sections (Page 13 and Page 21).

We agree that a single biological outcome in pre-clinical studies may not reflect overall animal health. However, if a certain biological outcome is known from human research to be part of the mechanistic/causal pathway related to the disease then this outcome may be considered an important surrogate for overall health. These biological outcomes in pre-clinical studies may also help to inform the exploration of them as predictive or prognostic variables in human clinical trials. We thank the reviewer for making this point. We have now discussed these points in the limitations section of the discussion (Page 23).

8) The impact of the study would be greatly improved if a meta-analysis was performed including a higher number of studies. More recent studies need to be included as well.

Please see responses to reviewer comment #1.References

1) Dave, C., McRae, A., Doxtator, E. *et al.* Comparison of freshly cultured versus freshly thawed (cryopreserved) mesenchymal stem cells in preclinical in vivo models of inflammation: a protocol for a preclinical systematic review and meta-analysis. *Syst Rev* 9**,** 188 (2020). https://doi.org/10.1186/s13643-020-01437-z.

2) Galleu A, Riffo-Vasquez Y, Trento C, Lomas C, Dolcetti L, Cheung TS, et al. Apoptosis in mesenchymal stromal cells induces in vivo recipient-mediated immunomodulation. Sci Transl Med 9, 416 (2017). Available from: https://www.science.org/doi/10.1126/scitranslmed.aam7828